# Re-evaluation of the psychometric properties of ATE following changes in euthanasia regulations in Spain

**Daniel Lerma-García**[1], **María Laura Parra-Fernandez**[2], **Cristina Romero-Blanco**[2], **Sandra Martínez-Rodriguez**[2], **María Dolores Onieva-Zafra**[2]*

**1** Department of Nursing, University of Illes Balears, Ibiza, Baleares, Spain, **2** Faculty of Nursing Ciudad Real, University of Castilla-La-Mancha, Ciudad Real, Spain

* mariadolores.onieva@uclm.es

## Abstract

Understanding healthcare professionals' attitudes toward euthanasia, especially within teams assisting patients who request it, is essential for providing appropriate support throughout the process. The objectives of this study were to re-adapt and validate the Attitude Towards Euthanasia Scale for the Spanish context, to examine its dimensional structure, and to estimate its reliability, particularly in light of the 2021 regulation of euthanasia in Spain. A cross-sectional study was conducted with a non-probabilistic sample of 778 healthcare professionals from the Balearic Health Service. Data collection involved a self-reported sociodemographic questionnaire and the Attitude Towards Euthanasia Scale. The scale's psychometric properties were assessed through reliability and validity tests, including confirmatory factor analysis. The Attitude Towards Euthanasia Scale demonstrated strong reliability, with a Cronbach's alpha of α = 0.889 and item homogeneity ranging from 0.66 to 0.78. Factor loadings were reported for four models, including a unidimensional model and models with two, three, and four factors. The two-factor model explained 64.51% of the variance, with a Kaiser-Meyer-Olkin (KMO) value of 0.873. Fit indices indicated good model fit: root mean square residual = 0.040, goodness-of-fit index = 0.960, adjusted goodness-of-fit index = 0.927, and root mean square error of approximation = 0.069. After performing a Parallel Analysis, two loading factor were found. The adapted version of the Attitude Towards Euthanasia Scale, aligned with the current societal and legislative context, is a valid and reliable tool for assessing healthcare professionals' attitudes toward euthanasia, with psychometric properties consistent with the original version.

**Data Availability Statement:** The data contains potentially sensitive and re-identifiable information. However, researchers who meet the criteria for

## Background

The term "euthanasia" refers, according to its etymology, to the "good death" as the central axis of the bioethical debate on the dignity of health care at the end of life [1–5]. The attitudes of nurses and other health professionals towards euthanasia are crucial in shaping the implementation and ethical considerations of end-of-life care [6]. Euthanasia, the practice of

access to confidential data, may send their request to share data to department of Nursing of University of Castilla-La Mancha, to which the corresponding and last author belongs. The contact details are as follows: Dep. EnfermeriayFisioterapia@uclm.es.

**Funding:** The author(s) received no specific funding for this work.

**Competing interests:** The authors have declared that no competing interests exist.

intentionally ending a life to relieve pain and suffering, remains a controversial and complex issue within the healthcare sector [7]. Research has shown that nurses often experience ethical dilemmas when dealing with euthanasia requests, balancing the principles of beneficence and non-maleficence with respect for patient autonomy [8]. Moreover, other Studies indicate that nurses' perspectives on euthanasia are influenced by a variety of factors including personal beliefs, cultural context, level of education, and professional experience [9–11]. In regions where euthanasia is legally permitted, such as Belgium and the Netherlands, nurses play an integral role in the euthanasia process, which includes assessing patient eligibility and providing emotional support [12–14]. However, the attitudes towards euthanasia among nurses can vary widely [15,16]. For instance, a study by De Bal et al. (2006) [17] revealed that nurses who had more experience with end-of-life care were more likely to support euthanasia under strict regulatory frameworks. Conversely, nurses with strong religious or moral convictions often opposed euthanasia regardless of legal provisions [18,19].

There is an ongoing debate on the decriminalization of euthanasia in Spain, which tends to resurface following high-profile media cases. Medical Assistance in Dying (MAID) encompasses both the euthanasia process and assisted suicide, serving as the focal point in the current bioethical debate on end-of-life ethical dilemmas. Since the approval of Spain's Organic Law 3/2021 regulating euthanasia, the number of patients requesting or receiving MAID has been steadily increasing [20–29]. Although this law was developed a few years ago, its implications continue to shape discussions in both medical and ethical fields [30]. On the other hand, there is also criticism in certain geographical areas where patients or relatives demand greater attention to these patients because health personnel are either unaware of the procedure or are conscientious objectors, i.e. by law they can choose not to get involved in the process and therefore not to provide the support, help and finally the action required in these cases where the will of the patient would be at the mercy of the attitude of the professionals in the face of euthanasia.

The new regulatory framework for the regulation of euthanasia in Spain [30], makes it necessary to assess the attitudes of health professionals towards euthanasia, since beliefs and attitudes may vary throughout life and also based on professional experience and training in aspects related to end-of-life care. The attitude manifested in the face of a person's wish to die can be motivated by beliefs that are rooted beyond the profession of caregiving and that, although they do not allow this attitude toward caregiving to be molded, they are determinant in the process of developing this aid in dying. The debate is itself endowed with this controversy, since even in a secular country like Spain, factors such as religious practice would be a determinant to be considered in the attitude towards euthanasia. In addition to religion, other factors such as gender, age, educational level, marital status or number of children have been observed as mediators of these attitudes in different studies carried out around the world [31–35].

Given the diversity of opinions and the significant role nurses play in end-of-life care, it is essential to develop reliable and valid instruments to assess their attitudes towards euthanasia. Such tools can help in understanding the underlying factors that shape these attitudes and guide the development of educational programs and policies that support ethical practice in euthanasia [36].

After the regulation of euthanasia in Spain, the report of the Deontological Commission of the College of Nursing of the Balearic Islands communicated the results of a self-developed survey in which the opinions of registered nurses were collected through an observational, descriptive and cross-sectional study, which showed that most of the nurses surveyed (86.6%) showed favorable attitudes towards the right of patients to decide when and how to die [37,38]. However, due to the limitations of a self-developed questionnaire to measure the

attitudes of professionals, validated questionnaires or measurement scales are required within the Health Sciences. There is consensus on the validity of these processes as long as they have an adequate methodological design that ensures their replicability in different social and healthcare contexts, based on a process of translation (not simply at the literary level), adaptation of the instrument to the context in which it will be applied and validation of the results of its application in terms of its psychometric properties. The introduction of the Euthanasia Law in Spain has significantly altered the social and healthcare context in which attitudes toward euthanasia are formed. Given these changes, it is crucial to reassess the Attitude Toward Euthanasia (ATE) scale to ensure that it remains relevant and accurately reflects current attitudes and perceptions among healthcare professionals. While the psychometric properties of the initial version of the ATE scale were acceptable, areas for improvement were identified. For instance, certain items may not adequately capture the nuances of contemporary attitudes shaped by recent legal and societal developments. This study aims to address these gaps by validating and adapting the ATE scale for a diverse sample of healthcare professionals in Spain. Therefore, the aim of this study has been to revise the first translation/validation of the Attitude Toward Euthanasia scale (ATE) by Fernandez et al. [39] using a larger and more heterogeneous sample of healthcare professionals. This approach will allow for a comprehensive understanding of attitudes across different roles within the healthcare sector. The study aims to identify the different dimensions of the questionnaire and to partially modify certain items to better align with the current cultural context in Spain, particularly in light of the regulation of euthanasia in 2021.Ultimately, the goal is to develop an instrument that will be assessed for its reliability and validity after the study is completed, reflecting the evolving attitudes of healthcare professionals toward euthanasia amidst recent legal changes.

## Material and methods

### Sample

All health professionals of the Balearic Health Service and specialist professionals in training in the Balearic Islands were invited to participate in the study. Convenience non-probabilistic sampling was used. The study did not consider probabilistic sampling as its primary objective was to achieve a large sample size. For the recruitment of the sample, once authorized by the different managements, through the respective research commissions, collaboration was requested for the dissemination of the questionnaire by global mailing to all the professionals of the different health services. The response rate for the study was 16.56%.

Data collection took place between from the1st of July to the 31st of November 2023, with a total of 778 responses. The average response time was 16:36 minutes for completion of the sociodemographic questionnaire and the ATE-ES-R scale. The questionnaire was completed anonymously and confidentially, without identification of personal data. Participants did not receive any financial incentive and the final study population consisted of 778 participants.

### Design

Based on the ATE scale [39,40], we conducted a new translation to enhance cultural and contextual applicability. The translation process involved two bilingual translators whose native language is Spanish. After the initial translation, a back-translation was performed by a professional translator whose native language is English. Subsequently, the research team reviewed the translated version and a second team, formed by members of the Health Care Ethics Committee of the Ibiza and Formentera Health Area, revised it to obtain a semantic and technical equivalence in appropriate bioethical language.

## Measurements

**Demographic information.** Participants provided self-reported sociodemographic characteristics, including age, gender, marital status, educational level, nuclear family status, and years of professional experience. Additionally, we included specific questions related to their knowledge of ethics and euthanasia. These questions aimed to assess their understanding of relevant laws and ethical dilemmas encountered in practice: (Do you know the national law regarding assistance in dying?, Have you attended any training on ethics?, Have you attended any training on euthanasia?, Do you consider yourself adequately trained on euthanasia?) among others.

**Attitude towards euthanasia scale.** The ATE scale, developed by Wasserman, Claire, and Ritchie in 2005[40]. It is a ten-item tool designed to assess attitudes towards euthanasia. A Cronbach's Alpha of 0.87 was reported. Responses are rated on a Likert scale from 1 to 5: (1) strongly disagree, (2) disagree, (3) undecided, (4) agree, and (5) strongly agree. Two items were reverse coded to check for response bias. Examples of items include: "If a patient in severe pain requests it, a doctor should remove life support and allow that patient to die" and "It is acceptable for a doctor to administer enough medication to a suffering patient to end their life if the doctor believes the patient's pain is too severe." The original factorial structure consisted of four dimensions: severe pain [items 1,3,9], no recovery [items 4,6], patient request [items 8,10], and doctor's authority [items 4,5,7]. Several items had cross-loadings with multiple dimensions. The scale was first translated and validated into Spanish by Fernández et al. in 2019, showing a Cronbach's alpha of 0.827 [39].

## Ethical approval and consent to participate

The research project report was approved by the Research Ethics Committee of the Balearic Islands, complying with the methodological, ethical and legal requirements set out in the 2013 Declaration of Helsinki with the code CEI: IB 5116/23 PI. The study included a Patient Information Sheet and a written Informed Consent Sheet, which were provided at the same time the questionnaire was completed. The data obtained are confidential and cannot be used for any purpose other than the aim of this study.

## Data analysis

For the statistical analyses, IBM SPSS AMOS version 26, Jamovi version 2.3, and jMetrik version 4.1.1 were utilized. Initially, data coding and exploration were conducted to prepare the dataset for analysis. Subsequently, a descriptive analysis of the variables was performed to characterize the sample composition. Descriptive statistics included frequency distributions with percentages for categorical variables, as well as means and standard deviations for quantitative variables. To assess the normality of quantitative variables, the Kolmogorov-Smirnov test (N>50) was utilized.

An Exploratory Factor Analysis (EFA) was conducted, utilizing the Kaiser-Meyer-Olkin (KMO) measure and Bartlett's test of sphericity to assess sample adequacy. The Pearson correlation matrix was employed, followed by principal component extraction and Varimax rotation with Kaiser normalization. Reliability was evaluated using Cronbach's $\alpha$ and Omega. After conducting the Exploratory Factor Analysis (EFA), we proceeded with a Confirmatory Factor Analysis (CFA) utilizing the maximum likelihood estimation method. The adequacy of the factorial solution was assessed through several fit indices, including chi-square ($X^2$), Root Mean Square of Residuals (RMSR), Root Mean Square Error of Approximation (RMSEA), Non-Normed Fit Index (NNFI), Comparative Fit Index (CFI), Goodness-of-Fit Index (GFI), and Adjusted Goodness-of-Fit Index (AGFI). An acceptable fit was indicated by an RMSR

value of 0.05 or lower. For RMSEA, a value below 0.05 was deemed indicative of a good fit, while values between 0.05 and 0.08 were classified as reasonable. Furthermore, NNFI and CFI values reaching 0.95 or above, along with GFI and AGFI values greater than 0.90, signified a well-fitting model. The AGFI should also surpass 0.90 to suggest optimal model adequacy. All fit indices were expected to fall within the range of 0 to 1, with a benchmark of 0.90. In addition, lower values for the standardized root mean square residual (RMR) and RMSEA were associated with a better fit, with a reference point set at 0.08. Additionally, a parallel analysis was executed to further validate the factor structure. The determination of the number of factors to retain was conducted using parallel analysis, and the reliability of the selected factors was assessed; confidence intervals at 95% were computed for both the item scores and the model metrics.

## Results

### Development and preliminary evaluation of the ATE-ES-R questionnaire

Prior to finalizing the ATE-ES-R questionnaire, we engaged in a multi-phase process to ensure its relevance and appropriateness for the Spanish context. In collaboration with the Health Care Ethics Committee, we modified the expression "the physician" to "the health care team" in all items, reflecting the collaborative nature of end-of-life care. While Law 3/2021 regulating euthanasia in Spain mentions 'consulting physician' and 'responsible physician,' it also emphasizes the involvement of the 'health care team,' explicitly acknowledging nursing professionals alongside physicians [30]. To establish preliminary face validity, we consulted 10 experts knowledgeable about the law and its applications in the context of euthanasia in the Balearic community. Following this consultation, we conducted a pilot study of the final draft of the ATE-ES-R questionnaire to evaluate its comprehension, suitability, and cultural appropriateness. During this phase, we identified unclear items and potential misinterpretations, ensuring the questionnaire was user-friendly. The feedback from professionals indicated no significant difficulties, leading to the final version of the ATE-ES-R questionnaire.

This multi-phase approach was essential to achieve semantic and technical equivalence, as today's recommendation emphasizes the importance of conducting cross-cultural adaptations of measurement tools. This ensures that linguistic aspects, cultural connotations, and variations are adequately addressed, which can significantly affect the feasibility of the tool.

### Sociodemographic characteristics

An analysis of the sociodemographic variables (refer to Table 1) reveals that the mean age of participants is 42.94 years, with a standard deviation of 10.37, indicating moderate variability in the age distribution. In terms of gender, a significant majority (75.8%) of the participants are women, reflecting a notable predominance of females in the nursing profession. Marital status indicates that nearly half of the respondents are married (46.8%), followed by single individuals (43.1%), separated or divorced (9.1%), and widowed (1.0%). Additionally, 53.5% of the participants have children, and 25.2% identify as religious.

Regarding professional engagement, 96.9% of respondents are actively employed, with the largest proportion falling within the 21 to 30 years of professional experience range. Educationally, 77.9% possess a university degree pertinent to their field, with nursing (46.4%) and medicine (21.5%) being the most frequently cited disciplines.

When addressing inquiries related to bioethics and euthanasia, a substantial 69.7% of professionals report having minimal or no knowledge of the legislation governing euthanasia, while 30.3% indicate a comprehensive understanding of the topic. Furthermore, only 25.3% of respondents consider themselves adequately trained in health care ethics, and a mere 31.7% acknowledge having received training specifically related to euthanasia.

**Table 1. Sociodemographic characteristics of the sample.**

| VARIABLES | N = 778 | M±DT |
|---|---|---|
| **Age** | | 42.94 +10.73 |
| | | **F (%)** |
| **Gender** | Male | 185(23.8%) |
| | Female | 589(75.8%) |
| **Country** | Spain | 713 (91.6%) |
| | Other countries | 65(8.3%) |
| | Single | 335 (45.2%) |
| | Married | 364 (46.8%) |
| | Divorced | 71 (9.1%) |
| | Widow | 8 (1.0%) |
| **Children** | Yes | 416 (53.5%) |
| | No | 362 (46.5%) |
| **Religiosity** | Yes | 196(25.2%) |
| | No | 582(74.8%) |
| **Professional in active service** | Nurse | 365 (46.9%) |
| | Doctor | 180 (23.2%) |
| | Physiotherapist | 14 (1.8%) |
| | Psychologist | 14 (1.8%) |
| | Pharmacist | 7 (0.9%) |
| | Dentist | 4 (0.5%) |
| | Social worker | (0.3%) |
| | Other | 186 (23.9%) |
| **Years of experience** | 0–5 years | 121 (15.6%) |
| | 5–10 years | 114 (14.7%) |
| | 11–15 years | 102 (13.1%) |
| | 16–20 years | 112 (14.4%) |
| | 21–30 years | 198 (25.4%) |
| | More than 30 years | 99 (12.7%) |
| **Are you familiar with the Law on the Provision of Assistance in Dying?** | None | 48 (6.3%) |
| | Little | 484 (63.4%) |
| | A lot | 181 (23.7%) |
| | Completely | 50 (6.6%) |
| **Have you attended any training on ethics?** | Yes | 389 (51.0%) |
| | No | 374 (49.0%) |
| **Do you consider yourself adequately trained in ethics?"** | Yes | 179 |
| | No | 584 (68.3%) |
| **Have you attended any training on euthanasia?** | Yes | 242 (31.7%) |
| | No | 521 (68.3%) |
| **Do you consider yourself adequately trained in euthanasia?** | Yes | 95 (12.5%) |
| | No | 668 (87.5%) |

## Exploratory factor analysis

To conduct the exploratory and confirmatory factor analyses, the sample was divided into two parts. To ensure that the two subsamples share equivalent levels of common variance we used the Solomon method with the FACTOR software, Release version 12.04.05 x 64 bits. The Communality Ratio Index obtained was 0.98. The KMO values for each sample were 0.870 for the

**Table 2. Descriptive statistics by item.**

|  | M | IC 95% | DT | As | Kr |
|---|---|---|---|---|---|
| ATE1 | 3.35 | 3.24–3.46 | 1.15 | -0.366 | -0.478 |
| ATE2 | 3.35 | 3.23–3.48 | 1.23 | -0.363 | -0.615 |
| ATE3 | 3.31 | 3.20–3.43 | 1.18 | -0.440 | -0.449 |
| ATE4 | 3.28 | 3.16–3.39 | 1.15 | -0.321 | -0.701 |
| ATE5 | 3.14 | 3.02–3.26 | 1.20 | -0.097 | -0.721 |
| ATE6 | 3.31 | 3.19–3.42 | 1.12 | -0.305 | -0.568 |
| ATE7 | 2.96 | 2.85–3.08 | 1.12 | -0.019 | -0.720 |
| ATE8 | 3.76 | 3.65–3.87 | 1.11 | -0.916 | 0.277 |
| ATE9 | 3.32 | 3.20–3.44 | 1.17 | -0.271 | -0.663 |
| ATE10 | 3.87 | 3.76–3.97 | 1.05 | -0.932 | 0.558 |

first sample and 0.853 for the second sample. The sample size for EFA was 389 and the sample size for CFA: 389 [41]. In the exploratory factor analysis (EFA), the Pearson correlation matrix was utilized after assessing the skewness and kurtosis of the items using Mardia's test (Mardia test p-value = 0.643) (See Table 2). This approach was chosen because the data met the basic assumptions required for Pearson correlation, including normality. The results of the analysis indicated that the KMO index was 0.861, suggesting a high adequacy of the sample for conducting a factor analysis, as values above 0.8 are considered very good. Additionally, Bartlett's test of sphericity was significant (p = 0.000), indicating that the correlations between the variables are appropriate for this type of analysis, confirming that the correlation matrix is not an identity matrix.

Regarding the total explained variance, the analysis identified two factors that together account for 64.51% of the total variance of the construct studied. The first factor explains 52.07% of the variance, while the second factor accounts for 12.43%. Other factors were not retained as they had eigenvalues less than 1.

Varimax rotation was applied to facilitate the interpretation of the factors, aiming for each item to be primarily associated with a single factor. In the rotated component matrix, Factor 1 groups items related to a more permissive attitude toward the administration of medication or the withdrawal of life support at the patient's request, as observed in items 1, 10, 3, and 8, which have high factor loadings (between 0.760 and 0.798). Conversely, Factor 2 comprises items that reflect a more restrictive stance against the ending of the patient's life by the healthcare team, as indicated by items 6 and 9, which show high loadings on this factor (0.799 and 0.747, respectively).(See Table 3).

## Parallel analysis

The results of the parallel analysis indicate a comparison between the eigenvalues of the actual dataset and those generated randomly from 1,000 simulated datasets, utilizing a sample of 778 cases and 10 variables. After performing a Parallel Analysis, two loading factor were found, based on the eigenvalues of the real data that exceed the 95th percentile of the randomly generated values indicating the number of significant components. Components are retained only if their eigenvalues from the actual data exceed the 95th percentile of the simulated data. In this instance, the first eigenvalue from the actual data (5.139) is greater than the 95th percentile of the first simulated eigenvalue (1.229). Additionally, the second eigenvalue from the actual data (1.179) surpasses the 95th percentile of the second simulated value (1.163). However, beginning with the third eigenvalue (0.872), the actual data values fall below the 95th percentile of the simulated data (1.116), indicating that only the first two components should be retained for further analysis. Therefore, the parallel analysis is able to identify only two factors (see Fig 1).

**Table 3. Rotated loading matrix of the ATE.**

| Rotated loading matrix of the ATE | Domain | |
|---|---|---|
| | **1** | **2** |
| 1. ATE Item 1 | .798 | |
| 10. ATE Item 10 | .787 | |
| 3. ATE Item 3 | .768 | |
| 8. ATE Item 8 | .760 | |
| 6. ATE Item 6 | | .799 |
| 9. ATE Item 9 | | .747 |
| 5. ATE Item 5 | | .694 |
| 2. ATE Item 2 | | .655 |
| 4. ATE Item 4 | | .619 |
| 7. ATE Item 7 | | .599 |

Extraction Method: Principal Component Analysis.

Rotation Method: Varimax with Kaiser Normalization.

## Confirmatory factor analysis

In the confirmatory factor analysis performed in this study (n = 389) and in addition to the two-factor model suggested by the EFA, one-, three-, and four-factor models were tested in the CFA. The one-factor model was included as a baseline to examine the plausibility of a single general dimension explaining all items. The three- and four-factor models were tested based on theoretical and empirical considerations from previous research on similar instruments (39,41,60), where multidimensional structures have been observed. Testing these models

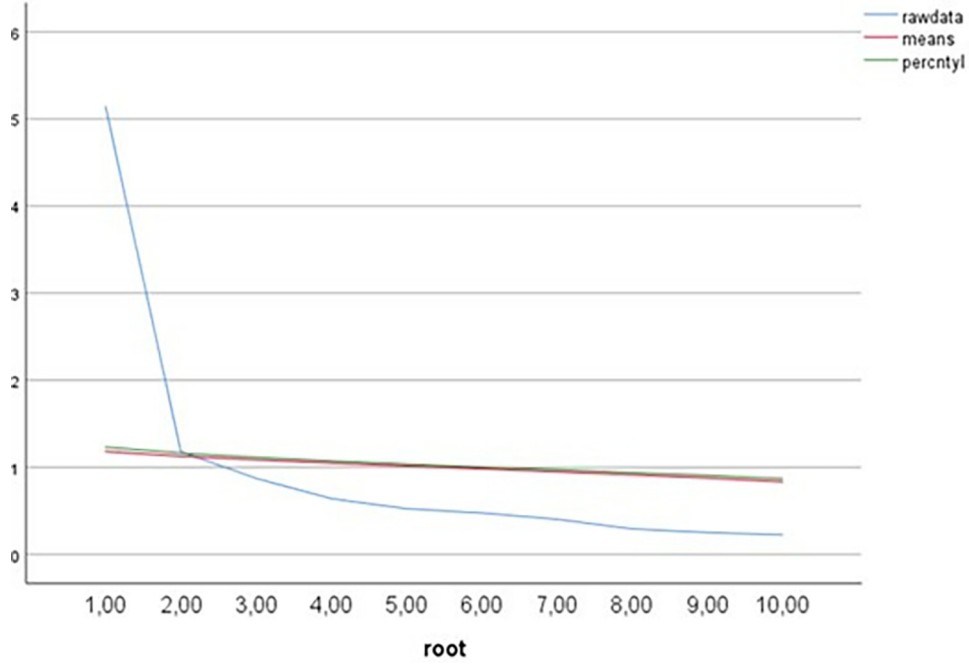

**Fig 1. Parallel analysis.**

**Table 4. Factorial loading solution for the four models.**

| | M1 | M2 | | M3 | | | M4 | | | |
|---|---|---|---|---|---|---|---|---|---|---|
| | F1 | F1 | F2 | F1 | F2 | F3 | F1 | F2 | F3 | F4 |
| ATE1 | 0.75 | 0.66 | - | - | 0.83 | - | - | 0.75 | - | - |
| ATE2 | 0.79 | - | 0.77 | 0.79 | - | - | 0.76 | - | - | - |
| ATE3 | 0.89 | 0.77 | - | - | 0.86 | - | - | 0.88 | - | - |
| ATE4 | 0.69 | - | 0.79 | 0.74 | - | - | 0.74 | - | - | - |
| ATE5 | 0.83 | - | 0.74 | 0.84 | - | - | 0.85 | - | - | - |
| ATE6 | 0.40 | - | 0.64 | - | - | 0.77 | - | - | - | 0.65 |
| ATE7 | 0.79 | - | 0.83 | 0.80 | - | - | 0.80 | - | - | - |
| ATE8 | 0.69 | 0.72 | - | - | 0.73 | - | - | - | 0.86 | - |
| ATE9 | 0.49 | - | 0.78 | - | - | 0.78 | - | - | - | 0.68 |
| ATE10 | 0.66 | 0.74 | - | - | 0.77 | - | - | - | 0.73 | - |

allowed for a comprehensive comparison to confirm that the two-factor solution provided the best fit. Four different models were constructed, each representing a different factor configuration for the subscales of the ATE questionnaire. The Kaiser-Meyer-Olkin (KMO) measure was 0.873, and Bartlett's test of sphericity was statistically significant (p = 0.000). Table 4 shows the loading on each item on the different factors in each model.

*Model 1 (M1): single factor model*

The analysis of the model reveals a limited fit, with a Chi-Square Minimum Discrepancy per Degree of Freedom (CMIN/DF) value of 4.139. The RMSEA is 0.090, suggesting an acceptable but not optimal fit, supported by a Probability Close Fit (PCLOSE) of 0.000. The RMR (Root Mean Residual) is 0.052, which is considered acceptable. The Akaike Information Criterion (AIC), with a value of 174.159, suggests a lower penalty for complexity, and the Expected Cross-Validation Index (ECVI) of 0.449 indicates that the model is not optimal. Although the model presents some acceptable fit indices, such as a Comparative Fit Index (CFI) of 0.950 and a Goodness of Fit Index (GFI) of 0.941, the overall fit is limited, as evidenced by an RMSEA of 0.090 and a CMIN/DF of 4.139. This suggests that, while the model has a reasonable foundation, there are areas that require improvement. The comparison with other models through the AIC and ECVI indicates that, while this model has advantages in terms of complexity, it is not the most optimal. (See Table 5 and Fig 2).

*Model 2 (M2): two factors*

The analysis of the CFA indicates that the sample is adequate, with a KMO value of 0.873 and a significant Bartlett's test of sphericity (p = 0.000). The chi-square ($\chi^2$) for the model is 85.080 with 30 degrees of freedom, resulting in a CMIN/DF of 2.836, suggesting an acceptable fit. The fit indices are positive, with an RMR of 0.040, a GFI of 0.960, and an AGFI of 0.927, all indicating good fit quality. Both the CFI and IFI are 0.971, and the NFI and TLI are close to 0.95, reinforcing the model's adequacy. The RMSEA is 0.069, within the acceptable range, and the PCLOSE value of 0.035 supports this interpretation. The AIC is 135.080, suggesting an appropriate balance between fit and complexity, while the ECVI of 0.348 indicates good fit for cross-validation. Finally, the Hoelter index shows that the model is robust and stable with larger sample sizes. (See Table 5 and Fig 2).

*Model 3 (M3): three factors*

The CFA for a three-factor model demonstrates a reasonable fit to the observed data. The chi-square value is 67.602 with 27 degrees of freedom, indicating differences between the observed and estimated matrices, but the CMIN/DF of 2.504 reflects an acceptable model fit. The RMSEA is 0.062, with a 90% confidence interval between 0.044 and 0.081, suggesting a

**Table 5. Fit indices for CFA models.**

|  | M1 | M2 | M3 | M4 |
|---|---|---|---|---|
| Chi-square ($\chi^2$) | 124.159 | 85.080 | 67.602 | 109.415 |
| Degrees of freedom (DF) | 30 | 30 | 27 | 29 |
| p-valor | 0.000 | 0.000 | 0.000 | 0.000 |
| CMIMN/DF | 4.139 | 2.836 | 2.504 | 3.773 |
| RMR | 0.052 | 0.040 | 0.034 | 0.039 |
| GFI | 0.941 | 0.960 | 0.968 | 0.964 |
| AGFI | 0.892 | 0.927 | 0.934 | 0.898 |
| CFI | 0.950 | 0.971 | 0.979 | 0.957 |
| IFI | 0.951 | 0.971 | 0.979 | 0.958 |
| NFI | 0.925 | 0.956 | 0.965 | 0.934 |
| TLI | 0.925 | 0.956 | 0.964 | 0.934 |
| RMSEA | 0.090 | 0.069 | 0.062 | 0.085 |
| PCLOSE | 0.000 | 0.035 | 0.129 | 0.000 |
| AIC | 174.159 | 135.080 | 123.602 | 161.415 |
| ECVI | 0.449 | 0.348 | 0.319 | 0.416 |

CMIN/DF (Chi-Square/df); RMR (Root Mean Square Residual); GFI (Goodness of Fit Index); AGFI (Adjusted Goodness of Fit Index); NFI (Normed Fit Index); IFI (Incremental Fit Index); TLI (Tucker-Lewis Index); CFI (Comparative Fit Index).

RMSEA (Root Mean Square Error of Approximation); AIC (Akaike Information Criterion); ECVI (Expected Cross-Validation Index).

good fit. The PCLOSE value of 0.129 indicates that the discrepancy is not significant. The RMR is 0.034, indicating low differences between observed and estimated covariances. The GFI is 0.968 and the AGFI is 0.934, both close to 1, which suggests a good fit. For comparative indices, the NFI is 0.965, IFI is 0.979, TLI is 0.964, and CFI is 0.979, all exceeding 0.90, indicating a good fit compared to the independent model. These indices support the notion that the proposed model adequately describes the data structure. The AIC for the model is 123.602, suggesting it is more parsimonious than the independence model but less so than the saturated model. The ECVI is 0.319, with a confidence interval between 0.266 and 0.391, indicating adequate performance for cross-validation in a new sample. (See Table 5 and Fig 2).

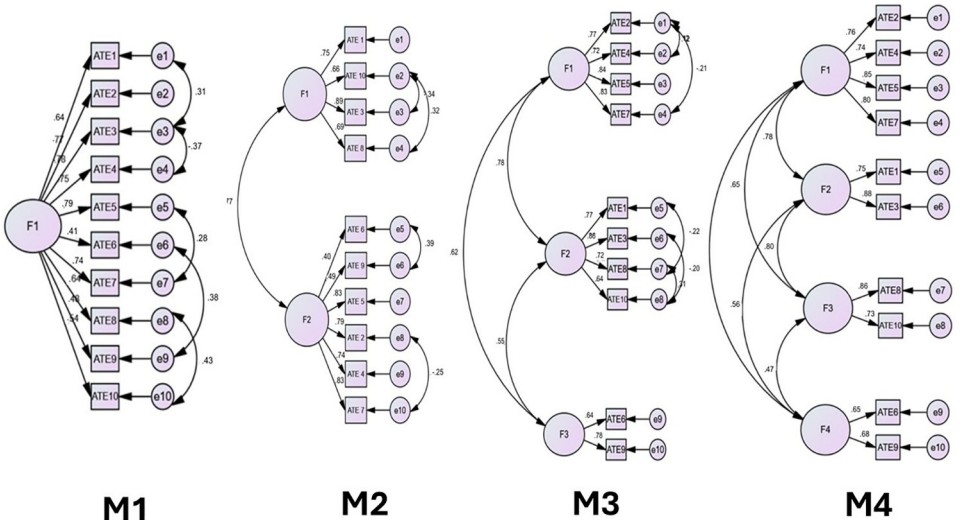

**Fig 2. CFA models.**

**Table 6. Cronbach's Alpha and McDonald's omega coefficients for measurement models.**

|  | M1 | M2 | | M3 | | | M4 | | | |
|---|---|---|---|---|---|---|---|---|---|---|
|  |  | F1 | F2 | F1 | F2 | F3 | F1 | F2 | F3 | F4 |
| α de Cronbach | 0.889 | 0.832 | 0.841 | 0.868 | 0.838 | 0.670 | 0.868 | 0.794 | 0.769 | 0.670 |
| ω de Mc Donald | 0.892 | 0.839 | 0.846 | 0.868 | 0.839 | 0.670 | 0.868 | 0.794 | 0.770 | 0.670 |

*Model 4 (M4): four factors*

The CMIN/DF value is 3.773, indicating inadequate fit. The is 0.085, with a 90% confidence interval ranging from 0.068 to 0.102 and a PCLOSE value of 0.000. This RMSEA suggests a moderate fit, as it exceeds the generally accepted threshold of 0.08 for good fit. The TLI is 0.934, which is close to the 0.95 threshold for good fit but does not reach it. The CFI is 0.957, indicating a good fit as it exceeds the 0.95 benchmark, reflecting a model that adequately explains the observed variance compared to an independent model. The GFI is 0.946, suggesting good fit, although slightly below the recommended value of 0.95. The AGFI is 0.898, just below the 0.90 threshold, indicating moderate fit. This suggests that while the model is adequate, it may not be parsimonious, as the AGFI adjusts the GFI based on the number of parameters. Finally, RMSR is 0.039, a low value indicative of good model fit. (See Table 5 and Fig 2).

## Reliability assessment

The reliability of the measurement instrument was evaluated using both Cronbach's Alpha and McDonald's Omega coefficients for all factors across the various models. (See Table 6) The results indicate that Model 1 (M1) and the two factors from Model 2 (M2) exhibited the highest reliability, with both Cronbach's Alpha and Omega values exceeding the acceptable threshold of 0.70. In contrast, Models 3 (M3) and 4 (M4) demonstrated factors with reliability coefficients significantly below 0.70, suggesting that these models may require further refinement to enhance the consistency of the items.

## Discussion

This study aims to revise the Attitude Toward Euthanasia scale (ATE) [41] using a larger and more diverse sample of healthcare professionals, particularly in light of recent changes in Spanish legislation regarding the legalization of euthanasia, to ensure the instrument's reliability and validity. After adapting the items of the questionnaire, the new ATE-ES-R version was obtained with a Cronbach's α of 0.892, which means a greater internal consistency with respect to the previous version (ATE-ES) of Fernández-Martínez [39], which was 0.827. This revision of the scale enhances the reliability of the instrument in Spanish, aligning it closely with the original Attitude Toward Euthanasia scale (ATE) developed by Wasserman [41], which has a Cronbach's α of 0.871, and the Persian version by Aghababaei [42], which boasts a Cronbach's α of 0.90.

After conducting statistical analyses, we identified four possible solutions for the factor structure of the ATE-ES-R questionnaire. However, the two-factor model (M2) proved to be the most robust and provided the best fit. The first factor, which we can refer to as 'patient request,' consists of items 1, 3, 8, and 10. This set of items begins with the expression 'If a patient,' which emphasizes the patient and their requests concerning the provision of aid in dying. The second factor could be termed "opinions on euthanasia" and is composed of items 2, 4, 5, 6, 7, and 9. In this dimension, we find items that begin with the expression "It is acceptable. . ." (2, 4, 5, and 7), guiding the respondent to express their position regarding the

practice of euthanasia. Additionally, there are items that start with the phrase "If a patient. . ." (6 and 9), which emphasizes the patient and their requests concerning assistance in dying, thereby providing a complete view of both professionals' and patients' perspectives on euthanasia.

The original Wasserman instrument [41] is structured on the basis of six dimensions in its initial design: "severe pain", "no recovery", "patient requests", "doctor's authority", "active euthanasia" and "passive euthanasia". However, in the confirmatory factor analysis, two factor loadings were obtained that explained up to 50% of the variance. Although the author did not list the items of each loading factor, the results of our model with two factors explain a higher percentage of variance (64.51%). Based on the theoretical dimensions of the original instrument, and in relation to our two-factor model, which is the most explanatory of the possibilities analyzed, Factor 1, 'patient requests,' encompasses all of Wasserman's theoretical dimensions except 'physician authority.' Similarly, Factor 2, 'opinions on euthanasia,' also includes all theoretical dimensions except 'patient requests.' The absence of the 'patient requests' dimension from the 'opinions on euthanasia' factor can be explained by considering that attitudes towards euthanasia do not depend on patient requests per se, but rather on each professional's ethical convictions, which precede the patient-care relationship [31–33,35,42–48].

It is interesting to note that factor 2, which includes items referring to patient requests for assistance in dying, excludes the authority of health professionals in decision-making. This reflects the fact that, although the process of requesting and providing assistance in dying requires the accompaniment of the patient's reference healthcare team, the final decision, in accordance with the regulations of each country, rests with the patient, who is at the center of healthcare. In this sense, it is thus the principle of autonomy, understood as Dworkin's "notion of the self which is to be respected, left unmanipulated, and which is, in certain ways, independent and self-determining" [49], which should ultimately frame the decision on the patient's request for euthanasia. This principle has been the subject of intense debate due to the controversy surrounding end-of-life dilemmas [50–57]. It can be argued, citing Kim (2021) [58] that "the more weight we put on autonomy, the more subjectively any eligibility requirement must be interpreted, so as not to offend autonomy". For this reason, the regulation of euthanasia allows the action to be carried out within the relationship between the patient and the care team, thus ensuring that this decision becomes a legitimate way to benefit the patient requesting it [59].

The fact of having changed the expression "the doctor" to "the health care team" may also have influenced in the disappearance of the "physician authority" dimension from the first factor. When analyzing the question posed to respondents about whether the patient's request for euthanasia should take precedence over the healthcare professional's conscientious objection, it becomes clear that the principle of autonomy acquires even more relevance, justifying the fact that the "doctor's authority" dimension does not appear in it. Among the different professions, physicians were the most reluctant to consider euthanasia over conscientious objection.

In the study by Aghababaei and Wasserman (2013) [60], the model of 2 loading factors is described as "voluntary euthanasia" and "non voluntary euthanasia". However, when we speak of the process of aid in dying based on the current bioethical consensus, these two terms are not used; rather, we refer to the MAID concept that includes euthanasia as a global concept without variants, and assisted suicide [18,61–63].

In the Spanish version of the EAS-ES by Fernández-Martínez et al. [30], three possible factor solutions are presented based on factor loadings with 1, 3, and 4 dimensions of the questionnaire. The three-factor loading model explains a percentage of variance of 52.6%, whereas in our study we present a percentage of variance for the 2-factor model of 64.51%. In their four-factor model, we can observe how the division of the resulting factor 1 from our proposal

("patient request") is split into two distinct factors: "severe pain" (items 1 and 3) and "patient request" (items 8 and 10). In this regard, the reasons that precede a patient's request for euthanasia (such as severe pain), in relation to their clinical condition, should not be a source of controversy, as these cases are already explained within the legislative framework [30]. Likewise, one can be either in favor or against, but it is not necessary to specify by dimensions the causes that precede a favorable or unfavorable attitude towards the application of euthanasia, and to continue the endless debate on positive or negative attitudes based on the specific cause [64–66]. We can consider that the mere presence of the item in the scale grants it a certain weight, which is accounted for in the overall factor and the scale as a whole, but not as distinct dimensions within a factor. The second factor ("opinions on euthanasia") is similarly fragmented into two different dimensions in the scale by Fernández-Martínez et al. [39] in their four-factor model: "no recovery" (items 6 and 9) and "doctor's authority" (items 2, 4, 5, and 7). In this case, we see how the acceptance of medical assistance in dying may be easier for healthcare professionals when identifying a patient who has no hope of restoring their optimal health status through medical interventions. On the other hand, it is important to consider that the questionnaire by Fernández-Martínez et al. [39] was validated prior to the regulation of euthanasia in Spain. We can observe how the items of the "doctor's authority" factor remain grouped within the same factor, reinforcing the idea that attitudes toward euthanasia, although they may be influenced by various circumstances, are rooted in the moral convictions of individuals and are often unchangeable [67–69]. Similarly, in the three-factor model by Fernández-Martínez et al. [39], items 6 and 9 were eliminated. The lack of weight of these items in this model can be explained by the fact that at the time of the questionnaire's validation, euthanasia was not regulated in Spain, meaning that professionals with negative attitudes toward euthanasia did not feel the need to express their opposition as strongly.

## Conclusions

The assessment of attitudes toward euthanasia among healthcare professionals operating within various health systems provides valuable information to gauge the level of controversy surrounding the regulatory processes of Medical Assistance in Dying (MAID) in different countries. When these regulatory processes are undertaken, measuring these attitudes using validated scales becomes an important component in identifying the needs of professionals involved in such requests. The ATE-ES-R version obtained is, therefore, one of these valid instruments for evaluating healthcare professionals' attitudes toward euthanasia, demonstrating psychometric characteristics similar to those reported by the original instrument. Given the internal consistency exhibited in a diverse sample of healthcare professionals and students in the Health Sciences (including undergraduate, postgraduate, or specialized training), this instrument may also be useful in the healthcare management related to the implementation of Law 3/2021 regulating euthanasia in Spain.

### Limitations

The first limitation of this study is the issue of self-report bias, where individuals may provide inaccurate or biased information about themselves. The second limitation is the social desirability bias, where respondents provide answers that they believe are socially acceptable or desirable rather than their true beliefs, especially on this sensitive topic. Additionally, the response rate was not very high. This could be due to the nature of the topic, which may be more challenging to address or provoke greater sensitivity. Moreover, the questionnaire was sent to work email addresses, specifically during work shifts when professionals typically use

their email. In this context, overloaded work schedules often make it difficult for professionals to find the time to respond to these surveys.

## Supporting information

**S1 File. ATE translation.**
(PDF)

## Acknowledgments

The authors want to thank all the study participants, all the Health worker of the Balearic Health Service for their participation in this study.

## Author Contributions

**Conceptualization:** Daniel Lerma-García, María Laura Parra-Fernandez, Cristina Romero-Blanco, María Dolores Onieva-Zafra.

**Data curation:** Daniel Lerma-García, Sandra Martínez-Rodriguez, María Dolores Onieva-Zafra.

**Formal analysis:** Daniel Lerma-García, Cristina Romero-Blanco, Sandra Martínez-Rodriguez, María Dolores Onieva-Zafra.

**Funding acquisition:** Daniel Lerma-García, María Laura Parra-Fernandez.

**Investigation:** Daniel Lerma-García, María Laura Parra-Fernandez, Cristina Romero-Blanco, María Dolores Onieva-Zafra.

**Methodology:** Daniel Lerma-García, Sandra Martínez-Rodriguez, María Dolores Onieva-Zafra.

**Project administration:** María Laura Parra-Fernandez, María Dolores Onieva-Zafra.

**Resources:** Daniel Lerma-García, María Laura Parra-Fernandez, Cristina Romero-Blanco.

**Software:** Daniel Lerma-García, Cristina Romero-Blanco.

**Supervision:** Daniel Lerma-García, María Dolores Onieva-Zafra.

**Validation:** Daniel Lerma-García, María Dolores Onieva-Zafra.

**Visualization:** Daniel Lerma-García, María Laura Parra-Fernandez, Cristina Romero-Blanco, Sandra Martínez-Rodriguez, María Dolores Onieva-Zafra.

**Writing – original draft:** Daniel Lerma-García, María Dolores Onieva-Zafra.

**Writing – review & editing:** Daniel Lerma-García, María Dolores Onieva-Zafra.

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
