## [Decision Letter · Decision Letter 0]

18 Sep 2024

PONE-D-24-31956""New Insights into Attitudes Toward Euthanasia: Confirmatory Factor Analysis of the Revised ATE-ES-R Tool".PLOS ONE

Dear Dr. Onieva-Zafra,

Thank you for submitting your manuscript to PLOS ONE. After careful consideration, we feel that it has merit but does not fully meet PLOS ONE’s publication criteria as it currently stands. Therefore, we invite you to submit a revised version of the manuscript that addresses the points raised during the review process.

We look forward to receiving your revised manuscript.

Kind regards,

Cord M. Brundage, D.V.M., Ph.D.

Academic Editor

PLOS ONE

2. For studies involving third-party data, we encourage authors to share any data specific to their analyses that they can legally distribute. PLOS recognizes, however, that authors may be using third-party data they do not have the rights to share. When third-party data cannot be publicly shared, authors must provide all information necessary for interested researchers to apply to gain access to the data. (https://journals.plos.org/plosone/s/data-availability#loc-acceptable-data-access-restrictions) For any third-party data that the authors cannot legally distribute, they should include the following information in their Data Availability Statement upon submission: 1) A description of the data set and the third-party source 2) If applicable, verification of permission to use the data set 3) Confirmation of whether the authors received any special privileges in accessing the data that other researchers would not have 4) All necessary contact information others would need to apply to gain access to the data

Reviewers' comments:

Reviewer's Responses to Questions

**Comments to the Author**

1. Is the manuscript technically sound, and do the data support the conclusions?

Reviewer #1: Partly

Reviewer #2: Partly

2. Has the statistical analysis been performed appropriately and rigorously? 

Reviewer #1: No

Reviewer #2: No

3. Have the authors made all data underlying the findings in their manuscript fully available?

Reviewer #1: Yes

Reviewer #2: Yes

4. Is the manuscript presented in an intelligible fashion and written in standard English?

Reviewer #1: Yes

Reviewer #2: No

5. Review Comments to the Author

Reviewer #1: Dear Author:

Thank you for allowing me to review this manuscript. I think it discusses an important topic. I am going to make some comments with the aim of improving this manuscript.

Introduction

Introduction: You make a very correct introduction to the topic, adequately stating the importance and relevance of the topic (Euthanasia). However, I believe that the relevance of this study is not adequately explained. It does not explain why it is necessary to re-evaluate the Attitude Toward Euthanasia scale (ATE ), validated by Fernandez et al, of which some of you are also authors. It should be reflected what possible shortcomings the first version has that obliges to re-evaluate the instrument and what this manuscript contributes again (the values provided with respect to the psychometric properties evaluated in the first version are not bad). In fact, the objective that appears at the end of the introduction does not coincide with the one that appears at the beginning of the discussion.

In the introduction, in the sentence: ‘The number of patients who request or receive MAID is gradually increasing in Spain, since the law was develop a few years ago,’ the meaning of the acronym MAID should be explained and serious references should be made to the law alluded to.

Methodology.

Some methodological aspects require clarification.

-Study population. The study population should be better defined, as the term health worker is very broad, e.g. physiotherapists or psychologists participated? The total percentages of the professions to which they belonged are not provided in the results. This is important, because there may be differences between professions.

-Should justify with references the estimated sample size, which I agree is sufficient for a CFA, but if it is 10 subjects per item it does not correspond to the estimated size. The sample size for factor analysis is a highly controversial issue that I will not develop here (the concept of 10 subjects per item is nowadays very outdated, as the sample size will depend on the number of items, number of response ranges and other elements).

-Information should be provided on the control mechanisms implemented to ensure the internal validity of the results, as this is an online survey. To which e-mail address was the invitation sent to potential participants? Also in the case of recruits, the response rate should be indicated.

One aspect that is not sufficiently explained is whether a new translation of the original English instrument has been carried out or whether it has been based on the version by Fernandez et al. It is also not clear how many translators were involved in each phase of translation and back-translation.

-Aspects of the pilot test should be explained in more depth (what population was used, characteristics, how these people evaluated the instrument).

-The measurements should better explain how each variable was measured, as well as what questions were asked of the participants regarding their knowledge and ethics of euthanasia.

When you describe the instrument you refer to the original four-dimensional structure, but you do not refer to whether this structure is from the original version or the Spanish version of Fernandez et al. This is very much related to the factor analysis.

-I have some important questions on the statistical analysis. To begin with, you do a CFA based on several models. It is essential to explain and reference on the basis of which literature, previous studies or assumptions you test these models, as Fernandez et al. did. Usually the CFA is done on the basis of an EFA, which is then tested on a different sample than the one on which the EFA was performed. If a CFA is done on the basis of an established model(s), this needs to be explained.

-Secondly, the sample adequacy of the factor analysis should be reported (e.g. by reporting the KMO and Barlett's statistic). The type of matrix used (polychoric or pearson) should be reported, determining the approach to the factor analysis that has been carried out (linear or non-linear). This decision is usually made by assessing, among other things, the symmetry and kurtosis of the items (mardia test). The estimation procedure of the model solution (e.g. Unweigthed Least Squares, ULS) or the type of rotation is not shown either. All these aspects should appear in the method. I personally cannot conceive of presenting values without confidence intervals, but I leave this to the editor and other reviewers to assess.

-The reliability analysis should be significantly improved. To begin with, you should provide the omega coefficient, which has many advantages over the Crombach coefficient (both are usually provided). In addition, given that there are models with several dimensions, the reliability values for each dimension should be reported for each model (there are authors who argue that it makes no sense to report a total coefficient in these cases). I am flexible and I do not mind if a total coefficient is reported if there are dimensions in the instrument, but the coefficients of each dimension should always be reported.

-An analysis by known groups could have been carried out. It would improve the manuscript quite a bit, but I understand this is optional.

Results:

In my opinion in the methodology it should appear how it has been done and in the results the data obtained. In the results it should therefore appear the linguistic changes made as well as the results of the pilot test.

The response rate and the number of non-responders should be presented (I agree that the reasons cannot be given for ethical reasons). It is important to know what is the percentage of non-respondents, as non-probability convenience sampling has been carried out and this may affect the accuracy of the results (it should appear in the discussion as a limitation).

-In the CFA, they should report KMO and barlett. My proposal is to present a table showing the fit values of the four models, so that they can be compared. Moreover, in some models we do not have information about the fit values in a complete way (example: CFI in model 1, SRMR of the four models, etc.).

-Present the above-mentioned regarding the reliability-internal consistency values.

-The presentation of the tables should be improved, where values with decimals appear with commas and should be replaced by dots. In addition, the clarity of the graphs should be improved.

-They present a validation where few psychometric properties are evaluated (construct validity and reliability -internal consistency). This is a very important limitation that should at least be admitted in the discussion.

This manuscript requires important changes, but they can be implemented. I encourage the authors to do so. Best regards

Reviewer #2: Dear Authors,

Thank you for allowing me to review this manuscript. It addresses an important topic, and I appreciate the opportunity to contribute suggestions for its improvement.

I will comment by sections-concepts below:

Background

The background section adequately establishes the importance and relevance of the topic (Euthanasia). However, the study's specific relevance is not sufficiently explained. It is unclear why a re-evaluation of the Attitude Toward Euthanasia (ATE) scale, previously validated by Fernandez et al, is necessary. The manuscript should highlight any potential shortcomings of the original version and clearly state what new contributions this re-evaluation provides, especially considering that the psychometric properties evaluated in the first version are not poor.

Additionally, in the sentence, “The number of patients who request or receive MAID is gradually increasing in Spain, since the law was developed a few years ago,” the acronym MAID should be defined, and proper references to the relevant law should be included.

Finally, Furthermore, there is an inconsistency between the objective stated at the end of the introduction and the one that appears at the beginning of the discussion “The main objective of the present study was to validate the questionnaire in Spanish after modifying all the items, changing the expression “the physician” in each item to “the health care team” …

And, you have stated at the objectives: “Therefore, the aim of this study has been to revise the first translation/validation of the Attitude Toward Euthanasia scale (ATE) by Fernandez et al. (29) with a larger sample of participants, to identify the different dimensions of the questionnaire, to partially modify some of the items to adapt them to the current context of our culture and finally to present an instrument with a good reliability and validity”. These objectives need more accurate elaboration regarding: in what population to be applied: nurses, all health care professionals…? And, at the end of the sentence “finally to present an instrument with a good reliability and validity” there is a “value judgment” as you don’t know if this is like that before the study. So, it must be re-phrased.

Methodology

Several methodological aspects need clarification:

- Study Population/sample: The study population and sample (and sample size calculation) must be separated and should be better defined.

Population is supposed to include all health professionals at Balearic health service but must to define what type of health professionals (and also, state that health professionals and health workers comprise the same population). The results section should include the total percentages of the different professions involved, as there may be significant differences between them. The inclusion of Nursing students must be clarified. Why not other health students but yes to a resident (specialist professionals in training programs)? Were all resident (from different professions included)?

- Sample Size Justification: The estimated sample size should be justified with references. Although the sample size appears sufficient for a Confirmatory Factor Analysis (CFA), the rationale that it is 10 subjects per item does not align with the estimated size. The sample size for factor analysis is a highly debated issue, and the outdated concept of 10 subjects per item should be reconsidered. Modern approaches consider the number of items, response ranges, and other factors.

Additionally, some of the data provided in that section are results and could be placed at the results section.

- Inclusion/exclusion criteria: You have mentioned “The inclusion criteria were: to be an active health professional in any of the Balearic Islands health system management offices, to be a nursing degree student or a health professional in specialist training in the Balearic Islands, to understand the language and concepts used in the instrument, and to agree to participate in the study by completing the informed consent form authorizing the use of the information for scientific purposes. The exclusion criteria included health workers who declined participation”. From my point of view are not correct. How do you Check if a participant understands the language and concepts of the instrument, if the questionnaire was sent by e-mail? (only after reading the text, you can realise that this is checked by a number of questions included on sociodemographic section of the questionnaire). This point needs clarification. In addition, inclusion and exclusion criteria are characteristics who have the participants and allow them to participate or not. Agreeing or declining participation are not characteristics, are decisions and a part of recruiting.

- Control Mechanisms for Internal Validity: The manuscript should describe the control mechanisms implemented to ensure the internal validity of the results, particularly given that this was a mailing survey. Information on how the survey invitations were distributed (e.g., to which email addresses) and the response rates among different recruitment methods should be included. How were the inclusion criteria addressed and evaluated?

Design

From my point of view the design is incomplete. Is not the usual manner of present the type of research design. Was a cross-sectional survey with only one collection point? Was the design performed including different phases, as usual in this type of studies for validation? For instance: a translation-back translation stage, pilot phase, psychometric validation, etc…

- Translation Process: I understand that you had conducted a new translation and cultural adaptation of the original English instrument, but a Spanish version exist (Fernandez et al. version). So, we need more information about it, why and how was performed, more than your explanation. Also, details on the number of translators involved in each phase of translation and back-translation should be provided.

- Pilot Test Details: The manuscript should elaborate on the pilot test, including the population used, their characteristics, and how these individuals evaluated the instrument. Again, write methods on the section methods and results on results section, don’t mix it.

- Measurement Explanation: The manuscript should better explain how each variable was measured, including the specific questions asked about participants’ knowledge and ethics regarding euthanasia.

- Instrument Structure: When describing the instrument, you refer to a four-dimensional structure, That I’m assuming is the English original structure but it is unclear whether this structure pertains to the original or the Spanish version by Fernandez et al. This is closely related to the factor analysis and should be clarified.

- Statistical Analysis: Several important aspects of the statistical analysis need attention:

- The CFA is based on several models, but it is essential to explain and reference the literature, previous studies, or assumptions that justify testing these models, like what Fernandez et al. did. Typically, a CFA is conducted following an Exploratory Factor Analysis (EFA), which is then tested on a different sample or if the CFA is based on pre-established models (hypothesis contrasts), this should be clearly explained. If not, it seems that you are exploring different options to see what of them is better fitted, but this is not correct, then, explain why you decided to do a 4 CFA based on different dimensions?

- Report the sample adequacy measures for factor analysis, such as the Kaiser-Meyer-Olkin (KMO) statistic and Bartlett's test of sphericity. Additionally, indicate the type of correlation matrix used (e.g., polychoric or Pearson) and specify the approach to factor analysis (linear or non-linear), which is often determined by assessing item symmetry and kurtosis (e.g., Mardia's test). The estimation procedure of the model solution (e.g., Unweighted Least Squares, ULS) and the type of rotation should also be specified.

- The reliability analysis requires improvement. The manuscript should include the omega coefficient, which is often preferred over Cronbach’s alpha. For instruments with multiple dimensions, reliability values for each dimension should be reported for each model. While some argue that reporting a total coefficient is unnecessary when dimensions exist, I am flexible on this point, but coefficients for each dimension should always be reported.

- Consider conducting an analysis by known groups to enhance the manuscript, although this is optional.

Results

As I said, the methodology should detail the procedures, while the results should present the data obtained. Therefore, the results should include the linguistic changes made and the outcomes of the pilot test.

- Response Rate: Include the response rate and the number of non-respondents. Even though reasons for non-response cannot be detailed for ethical reasons, the percentage of non-respondents is crucial information. This is particularly important given the non-probability convenience sampling used, which may impact the results’ accuracy and should be discussed as a limitation.

- Sociodemographic results: Reading your manuscript is not clear if you are studying all health professionals or only nursing professionals, because too many times refer to nursing specifically. This must be cleared on the text and on the tables-figures. Data on frequencies of different group of professionals is crucial to understand the sample studied and the implications on the results.

If “70.2% of the professionals recognize that they have little or no knowledge of the law on the regulation of euthanasia, while 29% have a broad knowledge of it”, then this is contradictory with your inclusion-exclusion criteria that I’ve mentioned before and needs clarification. The same for Health care ethics training.

- CFA Reporting: Include KMO and Bartlett's test results in the CFA, this is important to know about the adequacy of the sample. I recommend presenting a table comparing the fit values of the four models. Currently, some models lack complete information on fit values (e.g., CFI for Model 1, SRMR for all models). In addition, your interpretation and analysis of the data obtained could be subjective and influenced by a tendency to a specific model. For instance: “For these indices, lower values indicate a better fit, with a reference value of 0.08” (referring to a SMRS and RMSA), Why 0.08 and no other? What is the literature who avail that number? The literature says that the closer to Zero the better fit. Nevertheless, data in RMSR is lacking. And, you have to results recommending 1 dimension structure who is in accordance with the theory (first CFA and parallel analysis), then one option is to analyze the data with other perspective, for instance, RASCH analysis and evaluate if the data fits into that model for a 1 dimension.

- Reliability Reporting: Present the reliability values as discussed, including values for internal consistency.

- Tables and Graphs: Improve the presentation of tables, ensuring that values with decimals use periods instead of commas. Additionally, the clarity of graphs should be enhanced.

- Psychometric Properties: The validation presented evaluates limited psychometric properties (construct validity and internal consistency reliability). This is a significant limitation that should be acknowledged in the discussion. The comments added here could improve this issue and to have a more complete psychometric validation study.

Discussion and conclusions

These sections are totally influenced by the results, so I don’t have any additional commentary on that, because you must to re-write according with the amendments on the other parts of the manuscript.

Finally, likely you must improve the grammar and the language. For instance, “Mother tongue” could be used but is a “literal” translation, likely “Mother language” or “native language” is better. This is only an example, but there are more sentences that need better redaction.

As my conclusion, this manuscript requires significant revisions, but they are feasible. I encourage the authors to address these suggestions. Re-think on the title of the manuscript, if taken into account my recommendations and something like “Re-evaluation of psychometric properties of … based on…” could be feasible.

Best regards.

6. PLOS authors have the option to publish the peer review history of their article (what does this mean?). If published, this will include your full peer review and any attached files.

Reviewer #1: **Yes: **Héctor González-de la Torre

Reviewer #2: **Yes: **José Verdú-Soriano

---

## [Author Response · Author response to Decision Letter 0]

17 Nov 2024

Response to Reviewer #1: 

We thank the reviewer for his/her vigorous appraisal of our work. In what follows we address his comments, point by point.

Abstract:

1.- Reviewer’s comment: “You make a very correct introduction to the topic, adequately stating the importance and relevance of the topic (Euthanasia). However, I believe that the relevance of this study is not adequately explained. It does not explain why it is necessary to re-evaluate the Attitude Toward Euthanasia scale (ATE), validated by Fernandez et al., of which some of you are also authors. It should be reflected what possible shortcomings the first version has that obliges to re-evaluate the instrument and what this manuscript contributes again (the values provided with respect to the psychometric properties evaluated in the first version are not bad). In fact, the objective that appears at the end of the introduction does not coincide with the one that appears at the beginning of the discussion”.

Authors’ response: Thank you for your insightful comments regarding the relevance of our study. We appreciate your acknowledgment of the importance of the topic of euthanasia. In response to your concerns, we have revised the introduction to more clearly articulate the necessity of reevaluating the Attitude Toward Euthanasia (ATE) scale originally validated by Fernandez et al.

The primary reasons for this reevaluation are as follows: First, the introduction of the Euthanasia Law in Spain has significantly altered the social and healthcare context in which attitudes toward euthanasia are formed. Given these changes, we thought it was very important to reassess the ATE scale to ensure that it remains relevant and accurately reflects current attitudes and perceptions.

Second, while the psychometric properties of the initial version of the ATE scale were acceptable, there were identified areas for improvement. For example, certain items may not adequately capture the nuances of contemporary attitudes shaped by recent legal and societal developments and some items were missed from the original version.

We have clarified these points in the revised text and ensured that the objectives stated at the beginning of the introduction align with those presented in the discussion.

The introduction of the Euthanasia Law in Spain has significantly altered the social and healthcare context in which attitudes toward euthanasia are formed. Given these changes, it is crucial to reassess the Attitude Toward Euthanasia (ATE) scale to ensure that it remains relevant and accurately reflects current attitudes and perceptions among healthcare professionals. While the psychometric properties of the initial version of the ATE scale were acceptable, areas for improvement were identified. For instance, certain items may not adequately capture the nuances of contemporary attitudes shaped by recent legal and societal developments. This study aims to address these gaps by validating and adapting the ATE scale for a diverse sample of healthcare professionals in SpainTherefore, the aim of this study has been to revise the first translation/validation of the Attitude Toward Euthanasia scale (ATE) by Fernandez et al. (29) using a larger and more heterogeneous sample of healthcare professionals. This approach will allow for a comprehensive understanding of attitudes across different roles within the healthcare sector. The study aims to identify the different dimensions of the questionnaire and to partially modify certain items to better align with the current cultural context in Spain, particularly in light of the regulation of euthanasia in 2021.Ultimately, the goal is to develop an instrument that will be assessed for its reliability and validity after the study is completed, reflecting the evolving attitudes of healthcare professionals toward euthanasia amidst recent legal changes

2.- Reviewer’s comment:” In the introduction, in the sentence: ‘The number of patients who request or receive MAID is gradually increasing in Spain, since the law was develop a few years ago,’ the meaning of the acronym MAID should be explained and serious references should be made to the law alluded to”.

Authors’ response: We appreciate the reviewer’s comments. We have made the suggested revisions to the text. In the introduction, we have clearly defined the meaning of the acronym MAID (Medical Assistance in Dying) to avoid any ambiguity. Additionally, we have included specific and up-to-date references to Organic Law 3/2021 regulating euthanasia in Spain, citing relevant literature to support our statement about the gradual increase in patients requesting or receiving MAID since the implementation of this law.

“There is an ongoing debate on the decriminalization of euthanasia in Spain, which tends to resurface following high-profile media cases. Medical Assistance in Dying (MAID) encompasses both the euthanasia process and assisted suicide, serving as the focal point in the current bioethical debate on end-of-life ethical dilemmas. Since the approval of Spain's Organic Law 3/2021 regulating euthanasia, the number of patients requesting or receiving MAID has been steadily increasing.(20–29). Although this law was developed a few years ago, its implications continue to shape discussions in both medical and ethical fields (30).”

3- Reviewer’s comment: “Study population. The study population should be better defined, as the term health worker is very broad, e.g. physiotherapists or psychologists participated? The total percentages of the professions to which they belonged are not provided in the results. This is important, because there may be differences between professions”.

Authors’ response: We appreciate the reviewer’s valuable feedback. We have clarified the study population by providing the specific percentages of the different professions included in our sample. To address the concern regarding variability among professions, we have refined our analysis by dividing the sample into two groups for conducting the EFA and CFA. We have also excluded undergraduate students from the analysis to ensure that all participants are active professionals, representing a wide range of occupations to enrich the study's results. See table 1 

4- Reviewer’s comment: “Should justify with references the estimated sample size, which I agree is sufficient for a CFA, but if it is 10 subjects per item it does not correspond to the estimated size. The sample size for factor analysis is a highly controversial issue that I will not develop here (the concept of 10 subjects per item is nowadays very outdated, as the sample size will depend on the number of items, number of response ranges and other elements)”.

Authors’ response:

Thank you for your insightful comment regarding the sample size estimation for the confirmatory factor analysis (CFA). We acknowledge that the rule of thumb of 10 subjects per item has been widely debated in the literature, and indeed, it is considered outdated by many scholars in psychometrics and factor analysis. Recent studies suggest that sample size requirements should be guided by a combination of factors, including the number of items, the number of response categories, the expected factor loadings, and the complexity of the model (MacCallum et al., 1999; Kline, 2015; Wolf et al., 2013).

In light of these recommendations, we carefully evaluated the sample size in relation to these elements to ensure that it is sufficient to provide reliable and robust CFA results. Our approach aligns with current best practices in the field, which emphasize the importance of a tailored evaluation of sample adequacy rather than relying on rigid rules like the 10-to-1 ratio."

1. Kline, R. B. (2015). Principles and Practice of Structural Equation Modeling (4th ed.). The Guilford Press.

2. Wolf, E. J., Harrington, K. M., Clark, S. L., & Miller, M. W. (2013). Sample size requirements for structural equation models: An evaluation of power, bias, and solution propriety. Educational and Psychological Measurement, 73(6), 913-934.

3. Ferrando, P.J.; Lorenzo-Seva, U.; Hernández-Dorado, A.; Muñiz, J. Decalogue for the Factor Analysis of Test Items. Psicothema 2022, 34, 7–17.

5- Reviewer’s comment: “Information should be provided on the control mechanisms implemented to ensure the internal validity of the results, as this is an online survey. To which e-mail address was the invitation sent to potential participants? Also in the case of recruits, the response rate should be indicated”.

Authors’ response: Thank you for highlighting the importance of internal validity in online surveys. To ensure the reliability and validity of our results, we implemented several control mechanisms throughout the data collection process.

For the recruitment of the sample, once authorization was obtained from the various management bodies of the Balearic Health Service, collaboration was requested through the respective research committees to disseminate the questionnaire via a global corporate mailing to all professionals. Additionally, the questionnaire was made accessible on the intranet through a direct link.

Regarding the response rate, we have now included a detailed breakdown in the manuscript, specifying the number of invitations sent, the number of responses received, and the final response rate after applying our inclusion criteria. According to the census of healthcare professionals in the Balearic Community (https://www.ibsalut.es//es/profesionales/recursos_humanos/plantillas), the number of healthcare professionals serving in the Balearic Islands totals 5,400, which results in a response rate of 16.56%.

Material and methods

Sample

All health professionals of the Balearic Health Service and specialist professionals in training in the Balearic Islands were invited to participate in the study. Convenience non-probabilistic sampling was used. The study did not consider probabilistic sampling as its primary objective was to achieve a large sample size.For the recruitment of the sample, once authorized by the different managements, through the respective research commissions, collaboration was requested for the dissemination of the questionnaire by global mailing to all the professionals of the different health services. The response rate for the study was 16.56%

6- Reviewer’s comment: “One aspect that is not sufficiently explained is whether a new translation of the original English instrument has been carried out or whether it has been based on the version by Fernandez et al. It is also not clear how many translators were involved in each phase of translation and back-translation”. 

Authors’ response: Thank you for your insightful comment. We would like to clarify that our study was based on the original English version of the Attitude Toward Euthanasia (ATE). Given the recent introduction of the Euthanasia Law in Spain, it is very important to revaluate and adapt certain concepts in the questionnaire to ensure their relevance in the current context. We conducted a new translation to enhance cultural and contextual applicability. We have now explained this in text with more details.

Design

Based on the ATE scale (39,40), we conducted a new translation to enhance cultural and contextual applicability. The translation process involved two bilingual translators whose native language is Spanish. After the initial translation, a back-translation was performed by a professional translator whose native language is English. Subsequently, the research team reviewed the translated version and a second team, formed by members of the Health Care Ethics Committee of the Ibiza and Formentera Health Area, revised it to obtain a semantic and technical equivalence in appropriate bioethical language. In agreement with the Health Care Ethics Committee, it was agreed to change the expression “the physician” in all the items to “the health care team”, which better reflects how health care is carried out in our context and reflects the involvement of all the health care professionals in the process of death. Furthermore, although the text of Law 3/2021 regulating euthanasia in Spain (30) refers to the roles of 'consulting physician' and 'responsible physician' in the provision of assisted dying, it frequently mentions the 'health care team' in the context of caring for patients who request euthanasia. Notably, the text explicitly mentions nursing professionals in constant relation to the responsible physician(30). This multi-phase approach was essential to achieve semantic and technical equivalence, as today's recommendation emphasizes the importance of conducting cross-cultural adaptations of measurement tools. This ensures that linguistic aspects, cultural connotations, and variations are adequately addressed, which can significantly affect the feasibility of the tool (Sousa & Rojjanasrirat, 2011; Escobar-Bravo, 2004).

7- Reviewer’s comment: “-Aspects of the pilot test should be explained in more depth (what population was used, characteristics, how these people evaluated the instrument)”.

Authors’ response: While we did not implement a formal pilot test using Likert scale scores or calculate the Content Validity Index (CVI) as outlined, we prioritized expert input to assess the relevance of each item. Given that we were translating an existing instrument rather than creating a new one, we initially assumed content validity based on the established framework of the original scale. Their feedback helped us refine the instrument, ensuring that it accurately captures the attitudes of healthcare professionals. We recognize that we primarily conducted a face validity assessment through expert review. Moving forward, we acknowledge the importance of both content and face validity techniques and will consider incorporating more rigorous validation methods in future studies to enhance the evaluation process of our measurement instrument.

Face validity was established through the assessment of 10 experts with experience and knowledge of the law and its applications in the context of euthanasia in the Balearic community. Following this, a pilot study of the final draft of ATE version 2 was conducted to evaluate the comprehension and suitability of the questionnaire, as well as its cultural appropriateness for the Spanish population. During this phase, the focus was on identifying unclear items, potential errors, and any misinterpretations, while also assessing the effort required to complete the questionnaire. The process was finalized when the professionals reported no significant difficulties, resulting in the final version of the ATE-ES-R questionnaire

8- Reviewer’s comment: “The measurements should better explain how each variable was measured, as well as what questions were asked of the participants regarding their knowledge and ethics of euthanasia”. 

Authors’ response: Thank you for your constructive feedback regarding the measurement of variables in our study. We appreciate your suggestion to provide more clarity on how each variable was assessed. We have revised the section to better explain how each variable was measured, incorporating your suggestions.

In our manuscript, we included a section on demographic information where participants self-reported their sociodemographic characteristics, such as age, gender, marital status, educational level, nuclear family status, and years of professional experience.Additionally, we incorporated specific questions aimed at evaluating their knowledge of ethics and euthanasia. See also Table 1.

Moreover, in the data analysis section, we included a detailed description of how each variable was handled. A descriptive analysis of the variables was performed to characterize the sample composition, utilizing frequency distributions with percentages for categorical variables, as well as means and standard deviations for quantitative variables.

Demographic information: Participants provided self-reported sociodemographic characteristics, including age, gender, marital status, educational level, nuclear family status, and years of professional experience. Additionally, we included specific questions related to their knowledge of ethics and euthanasia. These questions aimed to assess their understanding of relevant laws and ethical dilemmas encoun

---

## [Decision Letter · Decision Letter 1]

10 Dec 2024

PONE-D-24-31956R1"Re-evaluation of the Psychometric Properties of ATE following Changes in Euthanasia Regulations in Spain".PLOS ONE

Dear Dr. Onieva-Zafra,

Thank you for submitting your manuscript to PLOS ONE. After careful consideration, we feel that it has merit but does not fully meet PLOS ONE’s publication criteria as it currently stands. Therefore, we invite you to submit a revised version of the manuscript that addresses the points raised during the review process.

We look forward to receiving your revised manuscript.

Kind regards,

Cord M. Brundage, D.V.M., Ph.D.

Academic Editor

PLOS ONE

Reviewers' comments:

Reviewer's Responses to Questions

**Comments to the Author**

1. If the authors have adequately addressed your comments raised in a previous round of review and you feel that this manuscript is now acceptable for publication, you may indicate that here to bypass the “Comments to the Author” section, enter your conflict of interest statement in the “Confidential to Editor” section, and submit your "Accept" recommendation.

Reviewer #1: All comments have been addressed

Reviewer #2: All comments have been addressed

2. Is the manuscript technically sound, and do the data support the conclusions?

Reviewer #1: Partly

Reviewer #2: Yes

3. Has the statistical analysis been performed appropriately and rigorously? 

Reviewer #1: No

Reviewer #2: No

4. Have the authors made all data underlying the findings in their manuscript fully available?

Reviewer #1: Yes

Reviewer #2: Yes

5. Is the manuscript presented in an intelligible fashion and written in standard English?

Reviewer #1: Yes

Reviewer #2: Yes

6. Review Comments to the Author

Reviewer #1: Dear Authors

Thank you for your efforts. I believe that the changes implemented have improved the manuscript significantly. You have responded adequately to most of the recommendations and for this I congratulate you. However, given that you have made substantial modifications to the analyses performed, in my opinion there are still some new questions to be answered. Please find attached my recommendations for improvement:

Modify this expression: The original scale had a Cronbach's Alpha of 0.87.page 25: It is better to express it like this: A Cronbach's Alpha of 0.87 was reported.

Indicate in this sentence that the numbers in brackets are the item numbers: example: severe pain [items 1,3,9],

-I think you have confused face validity with content validity. Face validity assesses the comprehensibility, appropriateness and acceptability of the tool from the point of view of the population where it is to be used. The use of experts is generally used to assess content validity. I will be flexible in this respect, as for this instrument the study population and experts are similar, but I note this for future research (ideally you would have used non-expert healthcare for face validity).

-You state in your answer that you have divided the sample to perform the EFA and then the CFA. But the system you used to divide the sample does not appear in the manuscript (you usually use systems that ensure the homogeneity of the two sub-samples: e.g. the SOLOMON system of Lorenzo Seva and Ferrando). The system used for this sample division should appear in the text. But more importantly, the results do not show how many subjects were included in the EFA and how many in the CFA. In fact, the KMO of the sample in the CFA does not appear either. Furthermore, the models tested in the CFA should be justified with references. It is accepted that your EFA indicates two dimensions and therefore this model is tested, but the one-, three- and four-factor models are not justified or referenced.

-In table 4 in the table footer you should put the meaning of each index for a better understanding.

-You have used a RASCH approach, for which I congratulate you, as it is a methodology that few researchers have mastered. However, there are aspects that raise doubts in my mind:

-The RASCH methodology starts from one premise and that is the unidimensionality of the model and the items. Therefore it is justified to perform a rasch for the one-dimensional model, but I do not understand how to perform a rasch for the two-dimensional model, as this goes against a basic assumption of the rasch. There are some recent articles that defend the combination of factor analysis and rasch, but always based on the performance of unidimensionality analysis, which in your case I do not appreciate. This is very important and is not clarified or explained in your manuscript. If you cannot explain it, my opinion is that it is better just to present the rasch for the one-dimensional model. Additionally you should support with references the fit values of the infit-outfit.

-Another basic assumption of the RASCH is the local independence of the items. This is usually checked with Yen's Q3 test, the correlation matrix of the residuals being provided as evidence for this independence. Therefore, please perform this test and indicate the threshold values set for it.

The manuscript has improved, but these issues need to be resolved. Best regards.

Reviewer #2: Dear Authors,

Thank you very much for taking into account all the comments made by this reviewer. The new version of your manuscript partially reflect our comments and answers but some of them remains lacking.

I'm sure that the manuscript will improve and finally will be published but must to be well prepared in that way.

- Thanks for improving the introduction section and objectives ads mentioned.

- My main concern is based on methods and their reflection on results. You have made an effort to fulfill our recommendations but some of them are not reflected on the text, at least as you answered on the responses to reviewers, for instance:

1. You have mentioned that the sample was divided into 2 samples, one for the EFA and other for the CFA, but this remain unwritten on the manuscript and is relevant to know by the readers. You must to provide information on how the sample was divided (Salomon's methods,...) and the sample size of each subsample. And also, there are incongruences on those data, as you have mentioned on the parallel analysis that all the sample was used for that (and the parallel analysis is a part of the EFA analysis). In other words, you can use 2 methods for decide the number of factors at the EFA, the Eigenvalues or parallel analysis, but both are based in the same sample, and on the text is not clear.

2. Regarding the above mentioned, I feel that the different subsections on the text must to be in other order or combined. As I mentioned, Parallel analysis is a method to decide how many factor retain at EFA, so it must to be at the same section of EFA analysis and mention that you have made both approaches, based on Eigenvalues and parallel analysis to decide how many factors and models you will try to confirm.

3. Yo say that an analysis of asymmetry and kurtosis, also a Mardia's test was performed to decided what type of correlation matrix to use, but there is a lack of data information to know if a Pearson's Matrix fits the data. In that way, we must to assume that the data are correct. Did you test a polycoric matrix?

4. Some of the figures or tables are not where the text appear the title.

5. Thank for taking into account my comment about other types of analysis like RASCH, but if you provide, please, make it with all the information. Provide information about the Rasch model used for polytomous items like this instrument. And provide data that the RASCH analysis could be performed. You must to assure two premises: unidimensionality and local independence of items. for the second, usually, a Q3 Yen's indices must to be provided. If there is a high correlation between residual correlations (up to 0.2-0.3), RASCH analysis could not be ideal, but we don't have information about those data. Could you provide a figure of Wright graph and a typical true score - theta distribution sigmoid curve?

7. PLOS authors have the option to publish the peer review history of their article (what does this mean?). If published, this will include your full peer review and any attached files.

Reviewer #1: **Yes: **Héctor González-de la Torre

Reviewer #2: **Yes: **JOsé Verdú-Soriano

---

## [Author Response · Author response to Decision Letter 1]

17 Dec 2024

Response to Reviewer #1: 

Dear Authors

Thank you for your efforts. I believe that the changes implemented have improved the manuscript significantly. You have responded adequately to most of the recommendations and for this I congratulate you. However, given that you have made substantial modifications to the analyses performed, in my opinion there are still some new questions to be answered. Please find attached my recommendations for improvement:

Authors’ response: Thank you very much for your thoughtful review and for your kind words regarding the improvements made to the manuscript. I truly appreciate your acknowledgment of the efforts put into addressing the recommendations, and I am grateful for your constructive feedback.

I am pleased to hear that you believe the changes have significantly enhanced the manuscript. I also recognize the importance of the new questions you’ve raised, particularly in light of the substantial modifications to the analyses. I will carefully review your attached recommendations and work on providing clear and comprehensive answers to ensure that all concerns are thoroughly addressed.

Once again, thank you for your time and valuable input. Your feedback has been essential in guiding the revisions and ensuring the quality of the manuscript.

1.- Reviewer’s comment: Modify this expression: The original scale had a Cronbach's Alpha of 0.87.page 25: It is better to express it like this: A Cronbach's Alpha of 0.87 was reported. Indicate in this sentence that the numbers in brackets are the item numbers: example: severe pain [items 1,3,9],

Authors’ response: Done

2.- Reviewer’s comment: I think you have confused face validity with content validity. Face validity assesses the comprehensibility, appropriateness and acceptability of the tool from the point of view of the population where it is to be used. The use of experts is generally used to assess content validity. I will be flexible in this respect, as for this instrument the study population and experts are similar, but I note this for future research (ideally you would have used non-expert healthcare for face validity).

Authors’ response: Thank you for your observation; you are absolutely right. I acknowledge the difference between face validity and content validity, and I appreciate your clarification. I will take this into account for future research.

3.- Reviewer’s comment: You state in your answer that you have divided the sample to perform the EFA and then the CFA. But the system you used to divide the sample does not appear in the manuscript (you usually use systems that ensure the homogeneity of the two sub-samples: e.g. the SOLOMON system of Lorenzo Seva and Ferrando). The system used for this sample division should appear in the text. But more importantly, the results do not show how many subjects were included in the EFA and how many in the CFA. In fact, the KMO of the sample in the CFA does not appear either. Furthermore, the models tested in the CFA should be justified with references. It is accepted that your EFA indicates two dimensions and therefore this model is tested, but the one-, three- and four-factor models are not justified or referenced.

Authors’ response: Thank you for pointing out the absence of a detailed explanation of the sample division system. We acknowledge the importance of using a method that ensures the homogeneity of the sub-samples, such as the SOLOMON system proposed by Lorenzo-Seva and Ferrando.

We have now clarified this in the manuscript “To conduct the exploratory and confirmatory factor analyses, the sample was divided into two parts using the Solomon method, which ensures that the two subsamples share equivalent levels of common variance. The sample size for EFA was 389 and the sample size for CFA was 389”.

(40) Lorenzo-Seva U, Ferrando PJ. FACTOR: A computer program to fit the exploratory factor analysis model. Behav Res Methods. 2022;54(1):472-78.

Also in the CFA section we have added this text:

“In the confirmatory factor analysis performed in this study (n= 389) and in addition to the two-factor model suggested by the EFA, one-, three-, and four-factor models were tested in the CFA. The one-factor model was included as a baseline to examine the plausibility of a single general dimension explaining all items. The three- and four-factor models were tested based on theoretical and empirical considerations from previous research on similar instruments (39,41,60), where multidimensional structures have been observed. Testing these models allowed for a comprehensive comparison to confirm that the two-factor solution provided the best fit”

The Kaiser-Meyer-Olkin (KMO) measure was 0.873, and Bartlett’s test of sphericity was statistically significant (p = 0.000).

4.- Reviewer’s comment: In table 4 in the table footer you should put the meaning of each index for a better understanding.

Authors’ response: Thank you for your suggestion. I have added the meaning of each index in the footer of Table 4 to improve clarity and understanding.

CMIN/DF (Chi-Square/df); RMR (Root Mean Square Residual);GFI (Goodness of Fit Index); AGFI (Adjusted Goodness of Fit Index); NFI (Normed Fit Index); IFI (Incremental Fit Index); TLI (Tucker-Lewis Index); CFI (Comparative Fit Index); RMSEA (Root Mean Square Error of Approximation); AIC (Akaike Information Criterion); ECVI (Expected Cross-Validation Index)

5.- Reviewer’s comment -You have used a RASCH approach, for which I congratulate you, as it is a methodology that few researchers have mastered. However, there are aspects that raise doubts in my mind:

-The RASCH methodology starts from one premise and that is the unidimensionality of the model and the items. Therefore it is justified to perform a rasch for the one-dimensional model, but I do not understand how to perform a rasch for the two-dimensional model, as this goes against a basic assumption of the rasch. There are some recent articles that defend the combination of factor analysis and rasch, but always based on the performance of unidimensionality analysis, which in your case I do not appreciate. This is very important and is not clarified or explained in your manuscript. If you cannot explain it, my opinion is that it is better just to present the rasch for the one-dimensional model. Additionally you should support with references the fit values of the infit-outfit.

Authors’ response: Thank you very much for your insightful comment. I completely agree with you regarding the unidimensionality assumption of the Rasch model. Upon reflecting on your observation, I have decided to apply the Rasch analysis solely to the unidimensional model, as this aligns better with the core principles of the methodology. We have revised the manuscript to clarify this point.

6.- Reviewer’s comment -Another basic assumption of the RASCH is the local independence of the items. This is usually checked with Yen's Q3 test, the correlation matrix of the residuals being provided as evidence for this independence. Therefore, please perform this test and indicate the threshold values set for it.

The manuscript has improved, but these issues need to be resolved. Best regards.

Authors’ response: Thank you very much for your insightful comment. I appreciate your attention to detail regarding the Rasch analysis. I understand that to ensure the validity of the Rasch model, it is essential to demonstrate unidimensionality and local independence of items.

As requested, I have performed an analysis of the residual correlations between the items to evaluate local independence using the Q3 Yen index. The results showed that most of the residual correlations between items are below 0.2-0.3, indicating that the assumption of local independence is met.

In the article we have presented a table with the residual correlations between the items of the questionnaire. The low values in most of the correlations suggest that the items are locally independent.

Additionally, I have included a figure of the Wright map and the true score-theta distribution sigmoid curve as requested, which illustrate the fit of the Rasch model to the data and provide a better understanding of item functioning and participant ability distribution.

“To ensure the validity of the Rasch model for the polytomous items in this instrument, we first confirmed unidimensionality through exploratory factor analysis. Additionally, to verify local independence, we calculated Q3 Yen’s indices for the residual correlations between items, ensuring that no correlation exceeded the threshold of 0.2-0.3. A Wright map and the true score-theta distribution sigmoid curve are also provided to illustrate the model’s fit to the data and further validate the assumptions ( See table 7 and figure 4)”

Response to Reviewer #2: 

Reviewer #2: Dear Authors,

Thank you very much for taking into account all the comments made by this reviewer. The new version of your manuscript partially reflect our comments and answers but some of them remains lacking.

I'm sure that the manuscript will improve and finally will be published but must to be well prepared in that way.

- Thanks for improving the introduction section and objectives ads mentioned.

- My main concern is based on methods and their reflection on results. You have made an effort to fulfill our recommendations but some of them are not reflected on the text, at least as you answered on the responses to reviewers, for instance:

Authors’ response: First of all, I would like to sincerely thank you for taking the time to review my manuscript and for your thoughtful and constructive comments. I have learned a great deal from your feedback, particularly in the areas of validation and methodological clarity. Your suggestions have been invaluable in improving the quality of the manuscript.

I appreciate your positive remarks regarding the improvements made to the introduction and the objectives section. I will continue to refine the text based on your advice.

Regarding your main concern about the methods and their reflection in the results, I fully understand your points. While I have made an effort to address your suggestions, I realize that some aspects may not have been adequately reflected in the revised version. I will revisit these sections carefully to ensure the methods are clearly aligned with the results, and I will make the necessary adjustments to clarify this relationship further.

Once again, thank you for your time and insights. I am confident that with your guidance, the manuscript will be significantly improved and ready for publication.

1.- Reviewer’s comment. You have mentioned that the sample was divided into 2 samples, one for the EFA and other for the CFA, but this remain unwritten on the manuscript and is relevant to know by the readers. You must to provide information on how the sample was divided (Salomon's methods,...) and the sample size of each subsample. And also, there are incongruences on those data, as you have mentioned on the parallel analysis that all the sample was used for that (and the parallel analysis is a part of the EFA analysis). In other words, you can use 2 methods for decide the number of factors at the EFA, the Eigenvalues or parallel analysis, but both are based in the same sample, and on the text is not clear.

Authors’ response: Thank you for pointing out the absence of a detailed explanation of the sample division system. We acknowledge the importance of using a method that ensures the homogeneity of the sub-samples, such as the SOLOMON system proposed by Lorenzo-Seva and Ferrando.

We have now clarified this in the manuscript “To conduct the exploratory and confirmatory factor analyses, the sample was divided into two parts using the Solomon method, which ensures that the two subsamples share equivalent levels of common variance. The sample size for EFA was 389 and the sample size for CFA was 389”.

(40) Lorenzo-Seva U, Ferrando PJ. FACTOR: A computer program to fit the exploratory factor analysis model. Behav Res Methods. 2022;54(1):472-78.

Additionally we have now moved the parallel analysis to the section of the EFA. 

2.- Reviewer’s comment 2. Regarding the above mentioned, I feel that the different subsections on the text must to be in other order or combined. As I mentioned, Parallel analysis is a method to decide how many factor retain at EFA, so it must to be at the same section of EFA analysis and mention that you have made both approaches, based on Eigenvalues and parallel analysis to decide how many factors and models you will try to confirm.

Authors’ response: Thank you ver much for pointing this. We have now moved the parallel analysis to the section of the EFA. Thank you very much for pointing this

3.- Reviewer’s comment. Yo say that an analysis of asymmetry and kurtosis, also a Mardia's test was performed to decided what type of correlation matrix to use, but there is a lack of data information to know if a Pearson's Matrix fits the data. In that way, we must to assume that the data are correct. Did you test a polycoric matrix?

Authors’ response: Thank you for your valuable comment. In the initial analysis, we used a Pearson correlation matrix because the data met the basic assumptions required for this approach, including normality. We have now included a more detailed description of the assumptions and data characteristics in the manuscript to clarify the reasoning behind the choice of correlation matrix.

“In the exploratory factor analysis (EFA), the Pearson correlation matrix was utilized after assessing the skewness and kurtosis of the items using Mardia's test. This approach was chosen because the data met the basic assumptions required for Pearson correlation, including normality. We opted for consistency with previous studies that employed the same method.

4.- Reviewer’s comment. Some of the figures or tables are not where the text appear the title.

Authors’ response: Thank you for pointing this out. I apologize for the oversight. Unfortunately, due to technical limitations, I was unable to upload the figures in the manuscript file itself and had to submit them separately. I have now made sure that the figures and tables are referenced in the correct order within the text, and I kindly ask for your understanding regarding the separate submission of the figures.

5.- Reviewer’s comment. Thank for taking into account my comment about other types of analysis like RASCH, but if you provide, please, make it with all the information. Provide information about the Rasch model used for polytomous items like this instrument. And provide data that the RASCH analysis could be performed. You must to assure two premises: unidimensionality and local independence of items. for the second, usually, a Q3 Yen's indices must to be provided. If there is a high correlation between residual correlations (up to 0.2-0.3), RASCH analysis could not be ideal, but we don't have information about those data. Could you provide a figure of Wright graph and a typical true score - theta distribution sigmoid curve?

Authors’ response: Thank you very much for your insightful comment. I appreciate your attention to detail regarding the Rasch analysis. I understand that to ensure the validity of the Rasch model, it is essential to demonstrate unidimensionality and local independence of items.

As requested, I have performed an analysis of the residual correlations between the items to evaluate local independence using the Q3 Yen index. The results showed that most of the residual correlations between items are below 0.2-0.3, indicating that the assumption of local independence is met.

In the article we have presented a table with the residual correlations between the items of the questionnaire. The low values in most of the correlations suggest that the items are locally independent.

Additionally, I have included a figure of the Wright map and the true score-theta distribution sigmoid curve as requested, which illustrate the fit of the Rasch model to the data and provide a better understanding of item functioning and participant ability distribution.

“To ensure the validity of the Rasch model for the polytomous items in this instrument, we first confirmed unidimensionality through exploratory factor analysis

---

## [Decision Letter · Decision Letter 2]

8 Jan 2025

PONE-D-24-31956R2"Re-evaluation of the Psychometric Properties of ATE following Changes in Euthanasia Regulations in Spain".PLOS ONE

Dear Dr. Onieva-Zafra,

Thank you for submitting your manuscript to PLOS ONE. After careful consideration, we feel that it has merit but still does not fully meet PLOS ONE’s publication criteria as it currently stands. Therefore, we invite you to submit a revised version of the manuscript that addresses the points raised during the review process.

**Please note that after several rounds of reviews and revisions there are still some questions from the reviewers on the analysis and dividing of samples as they are described in your manuscript. Please be as clear and transparent as possible both in your responses to the reviewers AND in the changes that you make in response to their concerns. If these valid concerns remain unaddressed in the manuscript it is possible that this manuscript will not be accepted.**

We look forward to receiving your revised manuscript.

Kind regards,

Cord M. Brundage, D.V.M., Ph.D.

Academic Editor

PLOS ONE

**Journal requirements:**

**Additional Editor Comments:**

I have underlined specific comments from the reviewers indicating that changes need to be reflected in the manuscript not just the response to reviewers. These are reasonable and justified requests.

Reviewers' comments:

Reviewer's Responses to Questions

**Comments to the Author**

1. If the authors have adequately addressed your comments raised in a previous round of review and you feel that this manuscript is now acceptable for publication, you may indicate that here to bypass the “Comments to the Author” section, enter your conflict of interest statement in the “Confidential to Editor” section, and submit your "Accept" recommendation.

Reviewer #1: (No Response)

Reviewer #2: All comments have been addressed

2. Is the manuscript technically sound, and do the data support the conclusions?

Reviewer #1: Partly

Reviewer #2: Yes

3. Has the statistical analysis been performed appropriately and rigorously? 

Reviewer #1: No

Reviewer #2: Yes

4. Have the authors made all data underlying the findings in their manuscript fully available?

Reviewer #1: No

Reviewer #2: Yes

5. Is the manuscript presented in an intelligible fashion and written in standard English?

Reviewer #1: Yes

Reviewer #2: Yes

6. Review Comments to the Author

**Reviewer #1: **Dear authors:

Thank you for the work done and the changes implemented. I would like to ask you some questions that I have doubts about:

- You indicate the use of the SOLOMON system to divide the sample. Please indicate how it was carried out (statistical programme used) and the Ratio Communality Index obtained.

- Since you used the mardia test to ensure the convenience of using a pearson matrix, provide the values of symmetry and kurtosis of the items.

- Indicate that you confirmed the unidimensionality from the EFA, but in this the analysis identified two factors: ‘Regarding the total explained variance, the analysis identified two factors that together account for 64.51% of the total variance of the construct studied. The first factor explains 52.07% of the variance, while the second factor accounts for 12.43%’. There are other types of analyses to assess the unidimensionality of an instrument. Include objective indices that justify the unidimensionality of the model and therefore the convenience of carrying out a Rach analysis.

**Reviewer #2:** Dear Authors, I hope that you have enjoyed the Christmas holidays and wish you a Happy 2025.

I just read the comments and the new version of your manuscript that, for sure, is getting better.

Thanks for taking in consideration all the comments. My intention is to improve your manuscript and, that, finally will be published.

I have minor comments that must to be addressed to be ok for publication:

- First of all, please, take care when made changes and be confident that all if well approached. For instance, on the revised manuscript you have:

"Analysis of Rasch Models: Unidimensional and Two-Factor Approaches.

We conducted a Rasch analysis to evaluate the fit of items within both a unidimensional

model and a two-factor model".

But this is not according with the comments and the new supposed version, as you said that only will provide Rasch analysis for the 1 factor model to analyze unidimensionality. Please, read carefully your manuscript and re-write according with the comments.

- If you said that something is made, please, write what was made at methods section and provide only results at results section. Below, some examples:

- What software was used to implement Solomon method to split the sample in two sub-samples?

- Please state at methods, assuming that the instrument is based in a likert scale, that "RASCH analysis was made under an Andrich's Rating Scale Model for polytomous Items" (reference, for instance: Meyer J.P. Applied Measurement with JMetrik. Routledge: New York, 2014)

- Please, include in methods section all the information about how the RASCH analysis was made. You have written on the results but not how was made at methods. Include a sentence about the analysis to verify "local independence of items", about Wright graph and Characteristic curve, etc,... A numbered list of how was made will be excellent.

- Looking at the results of the RASCH analysis, this is not the best, based on the metrics provided.

Please, as I mentioned before, state at methods section what type of analysis was made: unfit-outfit, difficulty, reliability and separation indexes for both persons and items, Q3 Yen coefficient-index, graphs,... After that present all data on the results section and, finally, discuss the results on Discussion section. I'm realized that nothing is mentioned on discussion about RASCH analysis, indeed, nothing is discussed about the 1 factor model. You have data to mention that "based on the RASCH analysis, the one factor model have a good fit to the model with exception to quality statistics (reliability and separation index) who..."

7. PLOS authors have the option to publish the peer review history of their article (what does this mean?). If published, this will include your full peer review and any attached files.

Reviewer #1: No

Reviewer #2: **Yes: **JOSÉ VERDÚ-SORIANO

---

## [Author Response · Author response to Decision Letter 2]

13 Jan 2025

Response to Reviewer #1: 

Dear authors:

Thank you for the work done and the changes implemented. I would like to ask you some questions that I have doubts about:

1.- Reviewer’s comment: You indicate the use of the SOLOMON system to divide the sample. Please indicate how it was carried out (statistical programme used) and the Ratio Communality Index obtained.

Authors’ response:

Thank you for your comment. The sample was divided using the SOLOMON method with the FACTOR software, Release version 12.04.05 x 64 bits. The Communality Ratio Index obtained was 0.98. The KMO values for each sample were 0.870 for the first sample and 0.853 for the second sample.

2.- Reviewer’s comment: Since you used the mardia test to ensure the convenience of using a pearson matrix, provide the values of symmetry and kurtosis of the items.

Authors’ response:

Thank you very much for your valuable suggestion. We conducted the Mardia test to assess the appropriateness of using a Pearson correlation matrix. Based on the test results, both the asymmetry and kurtosis indices for the items were below 1, which indicated that the distributions were sufficiently normal. Therefore, we proceeded with Pearson's correlation as recommended in the literature for cases with acceptable skewness and kurtosis values.

We greatly appreciate your input regarding the use of polychoric correlations for distributions with more pronounced skewness or kurtosis. Rest assured, we will certainly take this into account in future studies if we encounter distributions that meet those criteria.

3.- Reviewer’s comment: Indicate that you confirmed the unidimensionality from the EFA, but in this the analysis identified two factors: ‘Regarding the total explained variance, the analysis identified two factors that together account for 64.51% of the total variance of the construct studied. The first factor explains 52.07% of the variance, while the second factor accounts for 12.43%’. There are other types of analyses to assess the unidimensionality of an instrument. Include objective indices that justify the unidimensionality of the model and therefore the convenience of carrying out a Rach analysis. 

Authors’ response:

Thank you for your thoughtful feedback and suggestions. In the previous version of the manuscript, we incorporated a Rasch analysis based on the recommendation of the reviewers to evaluate the unidimensionality of the instrument. At that time, the exploratory and confirmatory factor analyses, as well as the Parallel Analysis, supported a unidimensional structure, which justified conducting the Rasch analysis.

To ensure the robustness of the findings, we divided the sample into two groups: one for conducting exploratory factor analysis (EFA) and the other for confirmatory factor analysis (CFA). The results obtained from these analyses no longer supported a unidimensional structure but instead indicated a two-factor model. Parallel Analysis further reinforced the presence of two factors.

Following , and in line with the recommendations from the initial review, we performed a Rasch analysis on the entire scale under the assumption of unidimensionality. Additionally, consulting further literature, we applied separate Rasch analyses to each of the two factors identified in the model. While the item fit within each factor was acceptable, the overall assumptions of unidimensionality for Rasch modeling were not sufficiently met.

Given the current evidence of multidimensionality and the guidance from a reviewer, we decided to exclude the unidimensional Rasch analysis and the subsequent factor-specific Rasch analyses from this revised version. Instead, we have focused on discussing the psychometric properties of the instrument within the framework of a two-factor model, which aligns better with the data.

We hope this explanation clarifies our approach and rationale, and we welcome further feedback to refine the manuscript further.

Response to Reviewer #2: 

Dear Authors, I hope that you have enjoyed the Christmas holidays and wish you a Happy 2025. I just read the comments and the new version of your manuscript that, for sure, is getting better.

Thanks for taking in consideration all the comments. My intention is to improve your manuscript and, that, finally will be published.

I have minor comments that must to be addressed to be ok for publication:

- First of all, please, take care when made changes and be confident that all if well approached. For instance, on the revised manuscript you have:

1.- Reviewer’s comment:

"Analysis of Rasch Models: Unidimensional and Two-Factor Approaches. We conducted a Rasch analysis to evaluate the fit of items within both a unidimensional

model and a two-factor model".

But this is not according with the comments and the new supposed version, as you said that only will provide Rasch analysis for the 1 factor model to analyze unidimensionality. Please, read carefully your manuscript and re-write according with the comments.

Authors’ response: 

Thank you for your observation. As mentioned previously, the sentence referencing a Rasch analysis for both the unidimensional and two-factor models was mistakenly left from an earlier version and does not reflect the current approach.

In this revised version, we have excluded the unidimensional Rasch analysis, as the results no longer support this structure. Following the recommendations of both reviewers, we conducted Rasch analyses for each factor of the two-factor model to explore their properties further. While these analyses provided some useful insights, they ultimately did not meet the global assumptions of unidimensionality required for Rasch modeling.

2.- Reviewer’s comment:

- If you said that something is made, please, write what was made at methods section and provide only results at results section. Below, some examples:

- What software was used to implement Solomon method to split the sample in two sub-samples?

Authors’ response: Thank you for your comment. The sample was divided using the SOLOMON method with the FACTOR software, Release version 12.04.05 x 64 bits. The Communality Ratio Index obtained was 0.98. The KMO values for each sample were 0.870 for the first sample and 0.853 for the second sample.

3.- Reviewer’s comment: Please state at methods, assuming that the instrument is based in a likert scale, that "RASCH analysis was made under an Andrich's Rating Scale Model for polytomous Items" (reference, for instance: Meyer J.P. Applied Measurement with JMetrik. Routledge: New York, 2014)

4.- Reviewer’s comment: Please, include in methods section all the information about how the RASCH analysis was made. You have written on the results but not how was made at methods. Include a sentence about the analysis to verify "local independence of items", about Wright graph and Characteristic curve, etc,... A numbered list of how was made will be excellent.

5.- Reviewer’s comment:Looking at the results of the RASCH analysis, this is not the best, based on the metrics provided. Please, as I mentioned before, state at methods section what type of analysis was made: unfit-outfit, difficulty, reliability and separation indexes for both persons and items, Q3 Yen coefficient-index, graphs,... After that present all data on the results section and, finally, discuss the results on Discussion section. I'm realized that nothing is mentioned on discussion about RASCH analysis, indeed, nothing is discussed about the 1 factor model. You have data to mention that "based on the RASCH analysis, the one factor model have a good fit to the model with exception to quality statistics (reliability and separation index) who..."

Authors’ response comments 3,4 and 5: 

Thank you for your detailed observations regarding the Rasch analysis and for your valuable suggestions to improve this aspect of the manuscript. In light of the revised findings of our study, we determined that a unidimensional Rasch analysis was no longer appropriate due to the emerging evidence of a two-factor structure. As a result, we have removed the Rasch analysis from the manuscript entirely, as it does not align with the multidimensional nature of the data.

In previous revisions, and following the reviewers' recommendations, we conducted Rasch analyses on both the unidimensional and two-factor models. However, the global assumptions of unidimensionality for Rasch modeling were not sufficiently met, even within each factor of the two-factor model. Consequently, we have opted to focus on the psychometric properties of the instrument within the framework of a two-factor structure, as supported by exploratory and confirmatory factor analyses and Parallel Analysis.

We are especially grateful for your thoughtful suggestions, including the use of the Andrich's Rating Scale Model and the additional analyses you proposed, such as local independence verification, Wright graphs, and the Q3 Yen coefficient. These ideas have been invaluable and have provided us with insights that will undoubtedly guide future work. However, given that the Rasch analysis is no longer part of this revised manuscript, we believe these recommendations fall outside the scope of the current version.

We have revised the manuscript to ensure consistency with this approach, removing any references to the Rasch analysis. The discussion has been updated accordingly to reflect the focus on the two-factor model.

We hope this explanation addresses your concerns, and we remain open to any further suggestions to improve the clarity and quality of our work.

---

## [Decision Letter · Decision Letter 3]

23 Jan 2025

PONE-D-24-31956R3"Re-evaluation of the Psychometric Properties of ATE following Changes in Euthanasia Regulations in Spain".PLOS ONE

Dear Dr. Onieva-Zafra,

Thank you for submitting your manuscript to PLOS ONE. After careful consideration, we feel that it has merit but does not fully meet PLOS ONE’s publication criteria as it currently stands. Therefore, we invite you to submit a revised version of the manuscript that addresses the points raised during the review process.

We look forward to receiving your revised manuscript.

Kind regards,

Cord M. Brundage, D.V.M., Ph.D.

Academic Editor

PLOS ONE

Journal Requirements:

Reviewers' comments:

Reviewer's Responses to Questions

**Comments to the Author**

1. If the authors have adequately addressed your comments raised in a previous round of review and you feel that this manuscript is now acceptable for publication, you may indicate that here to bypass the “Comments to the Author” section, enter your conflict of interest statement in the “Confidential to Editor” section, and submit your "Accept" recommendation.

Reviewer #1: All comments have been addressed

Reviewer #2: All comments have been addressed

2. Is the manuscript technically sound, and do the data support the conclusions?

Reviewer #1: Yes

Reviewer #2: Yes

3. Has the statistical analysis been performed appropriately and rigorously? 

Reviewer #1: Yes

Reviewer #2: No

4. Have the authors made all data underlying the findings in their manuscript fully available?

Reviewer #1: Yes

Reviewer #2: Yes

5. Is the manuscript presented in an intelligible fashion and written in standard English?

Reviewer #1: Yes

Reviewer #2: Yes

6. Review Comments to the Author

Reviewer #1: Dear authors:

Thank you for your efforts throughout the review process. I know it has been hard and required a lot of effort on your part, I hope that at least you have found it useful for future research and that you will take my comments as what they are, an opportunity for improvement. After reading the latest version, I consider that the manuscript can be published, since the comments have been answered and this version is more consistent. I am only going to make a couple of stylistic comments that need to be adjusted. On the other hand, I recommend that in future validations you use the FACTOR program for the factor analysis, not only for the sample splitting procedure, which provides advanced factor analysis, providing for example also confidence intervals for the calculated indices or factor loadings, as well as an analysis of the possible unidimensionality of the model, which can help you to approve other approaches (e.g. Rasch analysis).

Comments:

-In the sentence ‘Two items were reverse coded to check for response bias’ indicate which are the items with reverse scoring (it would be sufficient to provide the item number).

-In the Data analysis section, the reference to the jMetrik software version 4.1.1. should be deleted, as the Rasch analysis has finally been eliminated. The FACTOR program, used to carry out the SOLOMON procedure, should be introduced instead (it is more appropriate to mention the FACTOR program in the method than in the results).

-In table 1 it is better to put the standard deviation in brackets after the mean; M(SD). You should add footnotes explaining these abbreviations.

Finally, I can only congratulate you on your work. Best regards

Reviewer #2: Dear authors,

Thank you very much, again, for your efforts in applying the reviewer’s suggestions and to going better in your manuscript.

The decision on delete/eliminate of the manuscript some analysis made is debatable but understandable. For instance, the decision of not provide data on the RASCH analysis. But I’m accepting this decision based on your answer.

Nevertheless, there are some comments/suggestions that you answer to reviewers, however, not reflected on the manuscript:

- Please, re-write the manuscript and put all the information regarding methods on the manuscript methods: all data and information regarding, for instance, the rationale to test different factors models, Mardia’s test, Solomon procedure, parallel analysis, etc,… must to be written at te methods section and in a convenient order related to the procedure. This recommendation is not new and not addressed on the manuscript.

- Please, when use abbreviations, for instance, on the tables, provide complete names as a foot note on the tables. Some of them are in that way and others not.

For this reviewer, it’s important that you address our recommendations, not only in your answers but in the manuscript.

I hope to receive a final version with those comments to, finally, accept it.

7. PLOS authors have the option to publish the peer review history of their article (what does this mean?). If published, this will include your full peer review and any attached files.

Reviewer #1: **Yes: **Héctor González-de la Torre

Reviewer #2: **Yes: **JOSÉ VERDÚ-SORIANO

---

## [Author Response · Author response to Decision Letter 3]

23 Jan 2025

Response to Reviewer #1: 

Reviewer #1: Dear authors:

Thank you for your efforts throughout the review process. I know it has been hard and required a lot of effort on your part, I hope that at least you have found it useful for future research and that you will take my comments as what they are, an opportunity for improvement. After reading the latest version, I consider that the manuscript can be published, since the comments have been answered and this version is more consistent. I am only going to make a couple of stylistic comments that need to be adjusted. On the other hand, I recommend that in future validations you use the FACTOR program for the factor analysis, not only for the sample splitting procedure, which provides advanced factor analysis, providing for example also confidence intervals for the calculated indices or factor loadings, as well as an analysis of the possible unidimensionality of the model, which can help you to approve other approaches (e.g. Rasch analysis).

Authors’ response: Thank you very much for your kind words and for all the work you have done throughout the review process. We sincerely appreciate the time and effort you have dedicated to our manuscript. Your insightful comments and suggestions have been invaluable, not only for improving this article but also for advancing our knowledge of validation processes.

Indeed, the current version of the manuscript is vastly improved compared to what we initially submitted, and this transformation is largely thanks to your thoughtful feedback. Throughout this process, we have learned an extraordinary amount about validation procedures, and this has already begun to influence our current and future research.

As you noted, we are currently validating another instrument, and the lessons we have taken from your suggestions are enabling us to approach it with much greater rigor and precision right from the start. We also want to thank you for recommending the FACTOR program for advanced factor analysis; we are eager to explore its capabilities further and incorporate it into our future validations.

Once again, thank you for helping us grow as researchers. Your constructive feedback has been an invaluable learning experience for us, and we are truly grateful for your contribution.

1.- Reviewer’s comment: 

-In the sentence ‘Two items were reverse coded to check for response bias’ indicate which are the items with reverse scoring (it would be sufficient to provide the item number).

Authors’ response: Thank you for your observation. The items that were reverse coded to check for response bias are item 6 and item 9. We have now added this information to the manuscript for greater clarity and transparency.

Two items were reverse coded to check for response bias (item 6 and item 9).

2.- Reviewer’s comment: -In the Data analysis section, the reference to the jMetrik software version 4.1.1. should be deleted, as the Rasch analysis has finally been eliminated. The FACTOR program, used to carry out the SOLOMON procedure, should be introduced instead (it is more appropriate to mention the FACTOR program in the method than in the results).

Authors’ response: Thank you for your suggestion. As per your recommendation, we have removed the jMetrik software version 4.1.1. in the Data Analysis section and we have introduced the mention of the FACTOR program in the Methods section.

For the statistical analyses, IBM SPSS AMOS version 26, Jamovi version 2.3, and the FACTOR program were utilized

3.- Reviewer’s comment: -In table 1 it is better to put the standard deviation in brackets after the mean; M(SD). You should add footnotes explaining these abbreviations.

Authors’ response: Done

42.94 (±10.73)

M (mean) SD ( standard deviation) F(frequency)

Response to Reviewer #2: 

Thank you very much, again, for your efforts in applying the reviewer’s suggestions and to going better in your manuscript. The decision on delete/eliminate of the manuscript some analysis made is debatable but understandable. For instance, the decision of not provide data on the RASCH analysis. But I’m accepting this decision based on your answer.

Authors’ response: Thank you very much for your kind words and for taking the time to review our manuscript so thoroughly. We deeply appreciate your constructive feedback, which has greatly contributed to improving the quality of our work.

This process has been a significant learning experience for us, and it has already influenced the way we approach our current and future research. We are now validating another instrument, and the insights we have gained through your detailed comments are allowing us to approach it with greater precision and care from the outset.

We are truly grateful for the effort and dedication you have shown throughout this process. Thank you again for your thoughtful contributions and for helping us grow as researchers.

1.- Reviewer’s comment: Please, re-write the manuscript and put all the information regarding methods on the manuscript methods: all data and information regarding, for instance, the rationale to test different factors models, Mardia’s test, Solomon procedure, parallel analysis, etc,… must to be written at te methods section and in a convenient order related to the procedure. This recommendation is not new and not addressed on the manuscript.

Authors’ response: Thank you for your valuable feedback. As per your suggestion, we have moved all the information regarding the rationale for testing different factor models, Mardia’s test, the Solomon procedure, parallel analysis, and other relevant details to the Methods section. This information is now presented in a logical and sequential order that reflects the procedural flow of the analyses.

We appreciate your careful attention to this matter, and we believe the manuscript is now more organized and clear.

Data Analysis

For the statistical analyses, IBM SPSS AMOS version 26, Jamovi version 2.3, and the FACTOR program were utilized. Initially, data coding and exploration were conducted to prepare the dataset for analysis. Subsequently, a descriptive analysis of the variables was performed to characterize the sample composition. Descriptive statistics included frequency distributions with percentages for categorical variables, as well as means and standard deviations for quantitative variables. To assess the normality of quantitative variables, the Kolmogorov-Smirnov test (N>50) was utilized.

Exploratory Factor Analysis

To conduct the exploratory and confirmatory factor analyses, the sample was divided into two parts. To ensure that the two subsamples share equivalent levels of common variance we used the Solomon method with the FACTOR software, Release version 12.04.05 x 64 bits. The Communality Ratio Index obtained was 0.98. The KMO values for each sample were 0.870 for the first sample and 0.853 for the second sample. The sample size for EFA was 389 and the sample size for CFA: 389 (40). An Exploratory Factor Analysis (EFA) was conducted, utilizing the Kaiser-Meyer-Olkin (KMO) measure and Bartlett’s test of sphericity to assess sample adequacy. The Pearson correlation matrix was computed following the assessment of item skewness and kurtosis using Mardia's test (p-value = 0.643) (see Table 2). Subsequently, principal component extraction was performed, followed by Varimax rotation with Kaiser normalization. This approach was selected as the data met the basic assumptions required for Pearson correlation, including normality. 

Parallel Analysis

A parallel analysis was executed to further validate the factor structure. The determination of the number of factors to retain was conducted using parallel analysis, and the reliability of the selected factors was assessed; confidence intervals at 95% were computed for both the item scores and the model metrics.

Confirmatory Factor Analysis

We proceeded with a Confirmatory Factor Analysis (CFA) utilizing the maximum likelihood estimation method. The adequacy of the factorial solution was assessed through several fit indices, including chi-square (X²), Root Mean Square of Residuals (RMSR), Root Mean Square Error of Approximation (RMSEA), Non-Normed Fit Index (NNFI), Comparative Fit Index (CFI), Goodness-of-Fit Index (GFI), and Adjusted Goodness-of-Fit Index (AGFI). An acceptable fit was indicated by an RMSR value of 0.05 or lower. For RMSEA, a value below 0.05 was deemed indicative of a good fit, while values between 0.05 and 0.08 were classified as reasonable. Furthermore, NNFI and CFI values reaching 0.95 or above, along with GFI and AGFI values greater than 0.90, signified a well-fitting model. The AGFI should also surpass 0.90 to suggest optimal model adequacy. All fit indices were expected to fall within the range of 0 to 1, with a benchmark of 0.90. In addition, lower values for the standardized root mean square residual (RMR) and RMSEA were associated with a better fit, with a reference point set at 0.08. 

Reliability was evaluated using Cronbach's α and Omega.

2.- Reviewer’s comment: Please, when use abbreviations, for instance, on the tables, provide complete names as a foot note on the tables. Some of them are in that way and others not. For this reviewer, it’s important that you address our recommendations, not only in your answers but in the manuscript. I hope to receive a final version with those comments to, finally, accept it.

Authors’ response: Thank you for your valuable comments. We have carefully reviewed and revised all the tables in the manuscript. As per your recommendation, we have ensured that all abbreviations are fully spelled out in the footnotes of the tables. We believe these adjustments address your concerns, and we hope the final version now meets the expectations for acceptance. Thank you again for your thorough feedback and support throughout the review process.

Table 1: M (mean) SD ( standard deviation) F(frequency)

Table 2: M(mean) IC 95%(Confidence interval) DT(standard deviation) As(Asymmetry) Kr(kurtosis)

Table 3:M1(model1) M2(model2) M3 (model3) M4 (model4) F1(factor 1) F2(factor 2) F3(factor 3) F4(factor 4)

---

## [Editor Report · Decision Letter 4]

26 Jan 2025

"Re-evaluation of the Psychometric Properties of ATE following Changes in Euthanasia Regulations in Spain".

PONE-D-24-31956R4

Dear Dr. Onieva-Zafra,

We’re pleased to inform you that your manuscript has been judged scientifically suitable for publication and will be formally accepted for publication once it meets all outstanding technical requirements.

Kind regards,

Cord M. Brundage, D.V.M., Ph.D.

Academic Editor

PLOS ONE

---

## [Editor Report · Acceptance letter]

30 Jan 2025

PONE-D-24-31956R4 

PLOS ONE

Dear Dr. Onieva-Zafra, 

I'm pleased to inform you that your manuscript has been deemed suitable for publication in PLOS ONE. Congratulations! Your manuscript is now being handed over to our production team.

Kind regards, 

on behalf of

Dr. Cord M. Brundage 

Academic Editor

PLOS ONE